# Sexual size dimorphism in mammals is associated with changes in the size of gene families related to brain development

Benjamin Padilla-Morales [1] ✉, Alin P. Acuña-Alonzo [2], Huseyin Kilili [1], Atahualpa Castillo-Morales[3], Karina Díaz-Barba [1,2,4], Kathryn H. Maher[5], Laurie Fabian [1], Evangelos Mourkas[6], Tamás Székely[1], Martin-Alejandro Serrano-Meneses [7], Diego Cortez [8], Sergio Ancona[9] & Araxi O. Urrutia [1,2] ✉

In mammals, sexual size dimorphism often reflects the intensity of sexual selection, yet its connection to genomic evolution remains unexplored. Gene family size evolution can reflect shifts in the relative importance of different molecular functions. Here, we investigate the associate between brain development gene repertoire to sexual size dimorphism using 124 mammalian species. We reveal significant changes in gene family size associations with sexual size dimorphism. High levels of dimorphism correlate with an expansion of gene families enriched in olfactory sensory perception and a contraction of gene families associated with brain development functions, many of which exhibited particularly high expression in the human adult brain. These findings suggest a relationship between intense sexual selection and alterations in gene family size. These insights illustrate the complex interplay between sexual dimorphism, gene family size evolution, and their roles in mammalian brain development and function, offering a valuable understanding of mammalian genome evolution.

Exploring the intricate interplay between sexual selection, genome evolution, and the origins of sexually dimorphism continues revealing new frontiers in evolutionary genomics. Sexual selection, a prominent evolutionary force, drives the dynamics of unequal competition among individuals of the same sex, shaping their contributions to the next generation[1]. As a result, sexually dimorphic traits, encompassing morphological, behavioural, physiological and life history differences between sexes, serve as hallmarks of sexual selection[2–4]. One notable pattern in mammals is sexual size dimorphism[5], arising from selective pressures on body size that differentially impact one sex over the other[2,6].

In mammals and birds, male-biased sexual size dimorphism (male-biased SSD), where males are larger than females, frequently mirrors the strength of sexual selection acting upon males[7–12]. In other species, like the chinchilla (*Chinchilla lanigera*)[13], females are larger than males (female-biased SSD)[5,6]. The evolutionary processes that drive female-biased SSD remain unclear and seem to be driven by factors like fecundity selection dissociated from the influence of sexual selection[13–15].

Previous research has associated the evolution of various phenotypic traits with sexual size dimorphism in birds and mammals.

[1]Milner Centre for Evolution, Department of Life Sciences, University of Bath, Bath BA2 7AY, UK. [2]Instituto de Ecología, UNAM, Mexico city 04510, Mexico. [3]Cardiff University, School of Medicine, Cardiff CF10 3AT, UK. [4]Licenciatura en ciencias genómicas, UNAM, Cuernavaca 62210, México. [5]NERC Environmental Omics Facility, Ecology and Evolutionary Biology, School of Biosciences, University of Sheffield, Sheffield S10 2TN, UK. [6]Zoonosis Science Centre, Department of Medical Sciences, Uppsala University, Uppsala, Sweden. [7]Departamento de Ciencias Químico Biológicas, Universidad de las Américas Puebla, Sta. Catarina Mártir, San Andrés Cholula, Puebla 72810, México. [8]Centro de Ciencias Genómicas, UNAM, Cuernavaca 62210, México. [9]Instituto de Ecología, Departamento de Ecología Evolutiva, UNAM, México City 04510, México. ✉e-mail: benjaminpadillams@gmail.com; a.urrutia@bath.ac.uk

Species with polygynous mating systems[16,17] often exhibit higher male-biased SSD and higher species body mass, a pattern known as Rench's rule[16]. Also, ornaments (such as fancy plumes and skin patterns, exaggerated tails) and armaments (i.e., antlers, enlarged fangs and spurs) indicate male-male competition[18]. Additionally, several mammalian lineages show an inverse correlation between male-biased SSD and brain size[11,19]. There is also evidence that brain size is linked with sexual selection, with higher levels of sexual selection associated with smaller brain sizes in some lineages[11,20,21]. These associations highlight the potential role of sexual selection in shaping evolutionary trajectories. However, the complex relationship between sexual selection and other traits, especially brain size, remains understudied.

Efforts to unravel the molecular underpinnings of sexual dimorphism have mainly centred on gene expression patterns[22], revealing significant differences in gene expression between males and females. Such genes typically exhibit rapid evolution, particularly when displaying male-biased gene expression[23]. Hundreds of genes across mammalian species exhibit conserved sex-biased expression[24], with a subset of these genes consistently showing sex bias across the vertebrate phylogeny[25].

Given that sexual selection is one of the major driving forces in the emergence and maintenance of biodiversity, more research is needed to understand better the molecular paths between sexual selection and genome sequence evolution. Sexual selection may impact the evolutionary dynamics of genome structure, particularly the evolution of sex chromosomes[26]. Increased sexual selection is also associated with faster gene evolution and turnover in genes associated with various aspects of male reproduction[23,27-29]. Tickle and Urrutia have recognised gene duplication as a significant source of functional innovation in genomes[30]. For example, they consider gene duplication of developmental-related genes to have played a significant role in the evolution of several vertebrate features; a clear-cut example is the duplication of the Hox gene clusters[31]. Groups of genes originating from an ancestral single-copy gene form a gene family, which can expand through further gene duplications or contract through gene deletions. Gene family size can be very dynamic over time[32], and these variations can provide insights into changes in the relative functional relevance of molecular functions and the molecular basis of complex phenotypes[33-35]. Gene family expansion and contraction have essential roles influencing adaptive phenotypic diversity[36], as evidenced by the correlations with different biological functions[37-39], including the evolution of brain size and morphology[33,34]. However, the linkage between gene family size changes and SSD evolution requires further exploration[1].

Here, we use comparative genomics to analyse the gene family expansion and contraction in 124 mammalian species (Fig. 1) to uncover their associations with SSD and potential involvement in brain development and function. Using functional annotations from humans, we then identified temporal expression patterns of genes within SSD-associated gene families in the brain across prenatal and adult stages. Additionally, we examined the presence of sex-biased gene expression among these genes. Through this multi-layered approach, we offer insights into the genomic correlates of SSD, thus enhancing our understanding of the molecular mechanisms associated with sexual size dimorphism using mammals—one of the best-studied model systems—as study organisms.

## Results

### Significant associations between SSD and gene family size

We utilised a phylogenetic generalised least-square (PGLS) analysis, incorporating Benjamini–Hochberg correction[40], to assess the relationship between gene family size (as the dependent variable) and SSD, alongside log10-transformed average body mass (as independent variables). A total of 5425 gene families in 124 mammalian species were included in the analysis. The inclusion of body mass as a separate variable in the model is supported by an allometric relationship between body mass and SSD, termed Rensch's rule[16], which is found within our study species set ($r = 0.378$; $p < 0.001$; Supplementary Fig. 1).

Our analysis revealed significant associations between gene family size changes and SSD. Notably, the PGLS analysis with SSD corrected by log-transformed average body mass found a total of 340 SSD-associated gene families exhibiting statistically significant expansion ($p < 0.05$; effect size ranging from $r = 0.243$ to $0.674$; Fig. 2a), whereas 405 showed statistically significant contraction ($p < 0.05$; effect size ranging from $r = -0.243$ to $-0.625$; Fig. 2a). Furthermore, five gene families presented statistically significant contractions with log10-transformed average body mass corrected by SSD (effect size ranging from $r = -0.314$ to $-0.407$; Supplementary data 1). The only contracting gene family shared between log10-transformed average body mass and SSD (OG0000577) was associated with the homeobox genes (Entrez IDs: 30062, 84859, 30712), related to biological processes in development and spermatogenesis. In the following sections, we will address genes within SSD-associated gene families as SSD-associated genes.

As it has been suggested that SSD among species with larger females compared to males is influenced by factors other than sexual selection, we repeated the PGLS analysis after excluding the 18 species where females are larger than males. After removing these species, 297 out of 340 gene families remained under expansion and 345 out of 405 under contraction with SSD + log10-transformed average body mass (Supplementary data 1). These results suggest that excluding species exhibiting female size bias does not influence the observed association between gene family size and male size bias. Importantly, all these gene families were consistent with the sets associated with SSD when species with female size bias were included (Supplementary Data 1).

Previous studies have linked relative brain size with both SSD[11,19,41] and gene family expansions[33] during mammalian evolution. To investigate this link, we examined gene family size in 57 species, for which data on relative brain size and SSD were available. We identified 290 gene families significantly associated with SSD in a PGLS, including SSD corrected by relative brain size, of which 51 showed gene family expansion ($p < 0.05$; effect size ranging from $r = 0.383$ to $0.581$), and 239 showed contractions ($p < 0.05$; effect size ranging from $r = -0.382$ to $-0.680$; Fig. 2a). For relative brain size, 52 gene families presented significant shifts in gene family size in the PGLS corrected by SSD. Among these, 25 were significantly under expansion ($p < 0.05$; effect size ranging from $r = 0.420$ to $0.677$), and 27 were significantly contracted ($p < 0.05$; effect size ranging from $r = -0.430$ to $-0.612$; Supplementary data 1). There was minimal overlap between the gene families associated with SSD and brain size, with only one gene family showing a significant association with both traits ($p = 0.257$). This lack of significant overlap suggests that independent sets of genes influence these phenotypes.

### Enrichment of SSD-associated genes in brain and olfactory functions

Characterising the 340 SSD-associated expanded gene families found in the gene family expansion and contraction analysis, we observed 36 significantly enriched biological process annotations (Fig. 2b). Among these, the "sensory receptor of smell" category was significantly overrepresented ($p < 0.001$), comprising 6.77% of the gene families significantly associated with the gene family expansion analysis (Fig. 2b).

For the 405 contracted SSD-associated gene families, 168 functional annotation categories demonstrated significant enrichment (Fig. 2b). Most of the enriched categories were associated with various aspects of development, including cell development (spinal cord development, skin development, muscle organ development, animal organ morphogenesis), general development (multicellular organism development) and brain development. Brain development-related

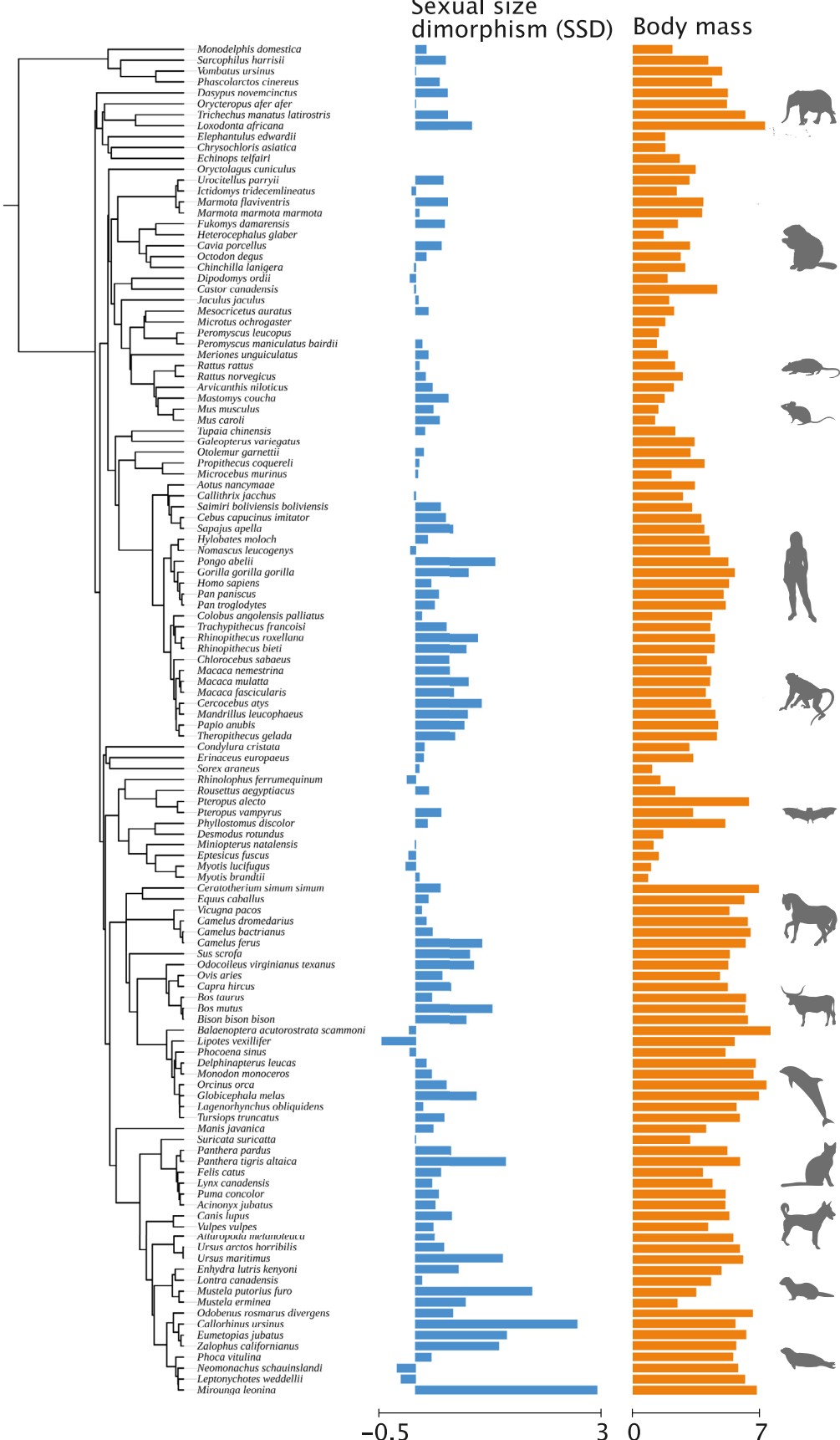

**Fig. 1 | Phylogenetic distribution of sexual size dimorphism and average adult body mass in mammals.** Blue bars represent sexual size dimorphism (log male body mass–log female body mass) data points, ranging from −0.5, where females are bigger than males, to 3, where males are bigger than females. Orange bars represent the logarithm of average body mass. The phylogeny encloses the 124 species assessed in this study.

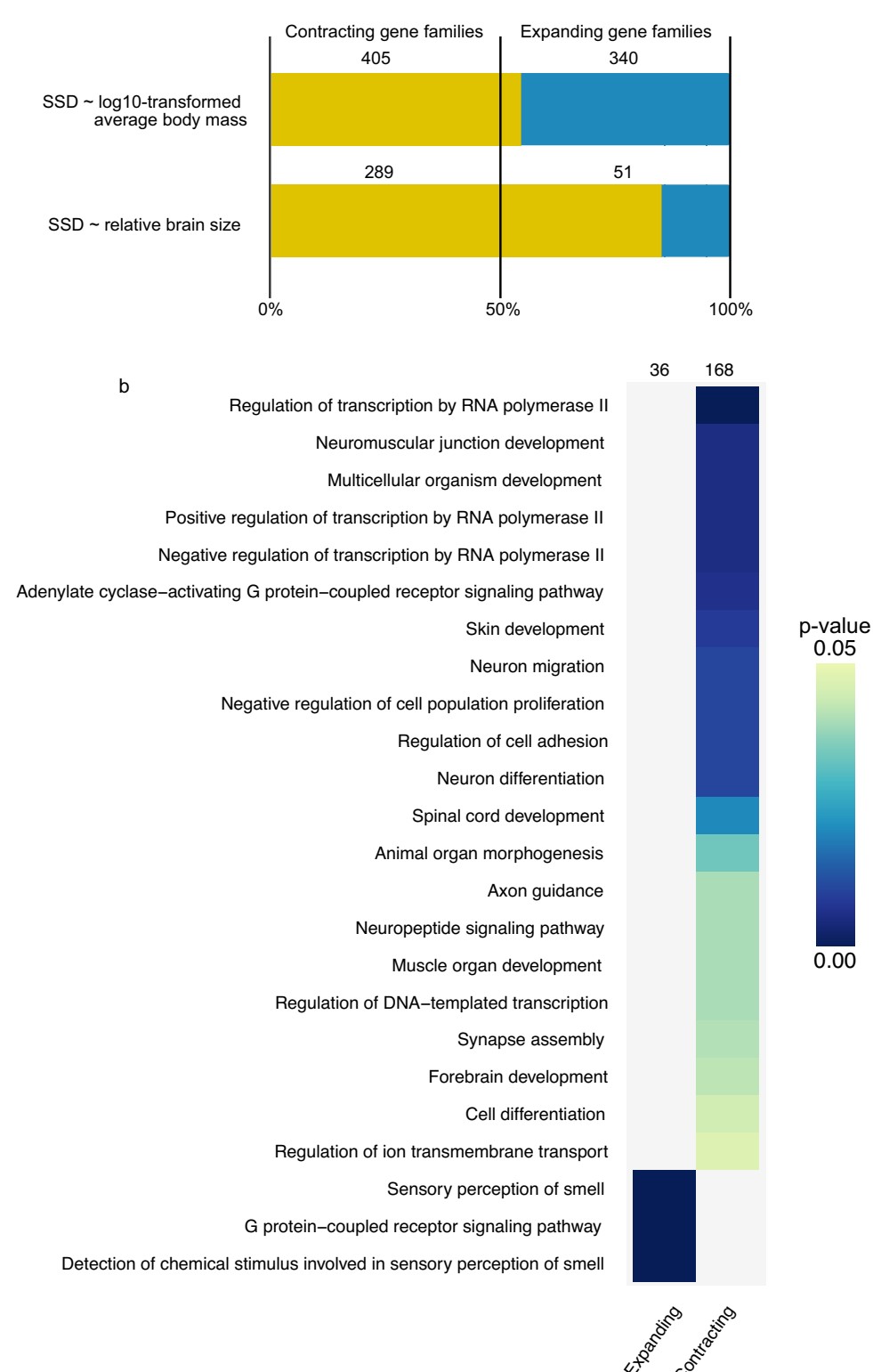

**Fig. 2 | Distribution of significantly associated and functional annotation overrepresenting expanding and contracting gene families in mammals.** Panels show the (**a**) number of contracting and expanding gene families associated with SSD; number of contracting and expanding gene families from two-predictor SSD corrected by body mass PGLS analysis; and number of gene families associated with SSD corrected by relative brain size. **b** Significantly enriched GO terms in heatmap are shown in colours ranging from light blue to dark blue. GO terms with smaller *p*-values are represented in darker colours. Individual columns from heatmap represent significantly enriched GO terms in expanding SSD-associated genes and contracting SSD-associated genes. The count of gene families associated with each category is indicated at the top of each column. Genes selected for GO enrichment analysis were retrieved from the SSD + log10-transformed average body mass gene family expansion and contraction analysis.

processes represented 8.75% of the total contracting families, with significant overrepresentation in neuromuscular junction development ($p = 0.001$), neuron migration ($p = 0.002$), differentiation ($p < 0.001$) and forebrain development ($p = 0.006$) (Fig. 2b).

## High gene expression of SSD-associated genes in adult brain

SSD-associated genes exhibited enrichment in brain development-related functions. If these genes support brain development, higher gene expression in the brain compared to other tissues is expected. Using human transcriptome expression profiles from 49 adult tissues and 20 from individuals in prenatal stages (Supplementary Data 2), we tested whether expanding and contracting SSD-associated genes are predominantly expressed in the brain. Overall, brain expression had the highest rank for expanding and contracting genes in adults and prenatal stages compared with other tissues (Supplementary Fig. 2a–d). Nonetheless, in adults, only the expression rank of contracting SSD-associated genes was significantly higher than expected by bootstrapping ($p < 0.001$; Supplementary Fig. 3b). For genes in prenatal stages, neither expanding nor contracting genes present an average rank for brain expression significantly higher compared to bootstrapped expectations ($p > 0.05$ in both datasets; Supplementary Fig. 3c, d). Then, after observing that SSD-associated genes in the brain exhibited strong temporal-dependent expression patterns in humans. We tested how genes are expressed through time using human transcriptomic data from the cortex, subcortex and cerebellum. Genes under expansion showed lower expression during prenatal stages and increased over time, while genes under contraction exhibited higher expression in prenatal stages that diminished in later stages (Fig. 3a, b).

## Sex bias in brain gene expression shows limited associations with SSD-associated genes

Since SSD is commonly associated with sex differences in gene expression[2,23,42], we assessed whether SSD-associated genes under expansion and contraction exhibit higher sex-biased expression in the human brain. Although there is a relatively high turnover of sex-biased genes[24], if SSD-associated genes tend to be sex-biased, we may expect a significantly higher degree of sex-biased gene expression among SSD-associated genes based on gene expression human data. Temporal patterns of cortical expression showed no difference in sex-biased gene expression ($p > 0.05$) in the cortex, subcortex and cerebellum (Supplementary Fig. 4a–d). For our comparative analysis of expression levels, we categorised all SSD-associated genes under expansion and contraction by GO categories related to the human brain. Then, we selected statistically significant gene expression log2-fold change values of each gene from prenatal individuals and adults. We found eleven statistically significant genes for SSD-associated genes under expansion from prenatal individuals, with most presenting a negative fold change. Still, the gene 3670 was the most downregulated (Entrez ID = 3670, GO categories: Visceral motor neuron differentiation, spinal cord motor neuron cell fate speciation, pituitary gland development, neuron fate specification, negative regulation of neuron differentiation; Fig. 3c). More than 25 statistically significant genes appear when contracting SSD-associated genes in prenatal individuals. Three of those genes are highly down-regulated (ENTREZ ID: 167826, 145258; log2 fold change < −2; Fig. 3d). We identified twelve statistically significant genes for SSD-associated gene under expansion in adults; however, none exhibited high up or down-regulation. Nonetheless, categories such as neuron differentiation, forebrain development, central nervous brain development and pituitary gland development categories arise (Fig. 3e). SSD-associated genes in adults presented more than 25 significant genes but only one with high overexpression (ENTREZ ID: 30062, GO category: hypothalamus development; log2 fold change < −2; Fig. 3f). For all the SSD-associated genes with significant fold change, refer to Supplementary Data 3.

## Discussion

Our results provide insights into mammals' intricate relationship between SSD and gene family size evolution. We identified significant associations between gene family expansions and contractions and SSD across 124 mammalian species through a comprehensive comparative genomics approach. Importantly, these expansions and contractions in gene family size are not explained as a by-product of gene family size variations related to the evolution of species' body mass, which is known to be closely correlated with SSD in mammals[16,43–45].

It is commonly accepted that in mammals, male-biased SSD is primarily driven by sexual selection[46–50] acting on males, while the role of sexual selection in the evolution of female-biased SSD has been more debated, though not entirely ruled out, unlike female-biased SSD, which is commonly attributed to fecundity selection[13,47,51]. The findings remained predominantly unaltered in analyses excluding 18 species, in which female size surpassed that of males. This suggests that interspecies male-biased sexual selection pressures potentially influence the observed correlations. Further studies focusing on clades like birds, which exhibit a wide range of size dimorphism and have a well-represented collection of sequenced genomes[52], are needed to understand better the evolution of genomic signatures linked to female-biased SSD.

Our results align with the well-established notion that SSD is often strongly associated with overall species' body mass[16,43,45]. (Supplementary Fig. 1). Notably, our gene family expansion and contraction studies mark one significantly associated gene family with both phenotypes. In humans, this gene family is represented by three genes (ENTREZ ID: 30062, 84839, 80712) involved in retinal and placental development and spermatogenesis[53–55]. Spermatogenesis genes commonly present rapid evolution, suggesting that sexual selection plays a role in molecular evolution[56], supporting our findings linking SSD and genome evolution.

Previous studies have shown that olfactory signals play an important role in mate choice[57,58], and in mice, sexual dimorphism in olfactory perception[59]. Our findings show that gene families that have significantly expanded in line with SSD are enriched in functional terms associated with olfactory functions. Within those, seven genes are key to smell perception (ENTREZ ID: 26532, 14627, 14918, 18249, 18575, 68795, 235380). It would be of interest to study how olfactory traits and sexual dimorphism correlate among other mammals.

SSD-associated gene families under contraction exhibit a significant overrepresentation of several biological functions related to brain function and development. We found 11 genes (ENTREZ ID: 5414, 7280, 9968, 10939, 11141, 17762, 23129, 27185, 57555, 64845, 171026) important for maintaining correct brain function and development. Defects, mutations, or deletions in any of these genes have been associated with various conditions, including mental disorders, syndromes like Opitz-Kaveggia, mental disability, and degenerative disorders[60–70]. To further explore the role of these gene families, we performed a PGLS including SSD and relative brain size. Only one gene family of crystalline genes was associated with both phenotypes, which included genes encoding chaperone proteins[71], involved in multiple functions including regulation of apoptotic processes, protein folding, muscle contraction[72,73], neural inflammation processes[74], as well as potential roles preventing the formation of toxic alpha-synuclein aggregation intermediates for Parkinson's disease[75]. Notably, crystalline alpha is overexpressed in several neurodegenerative disorders, such as Alexander's disease[76] and Alzheimer's disease[77]. These findings suggest a potential association between reduction of genes involved in brain-related functions and an increased vulnerability to brain disorders in species with high SSD.

We explored gene activity profiles for SSD-associated genes, focusing on humans as our representative species due to the extensive availability of transcriptomic resources across diverse tissues and

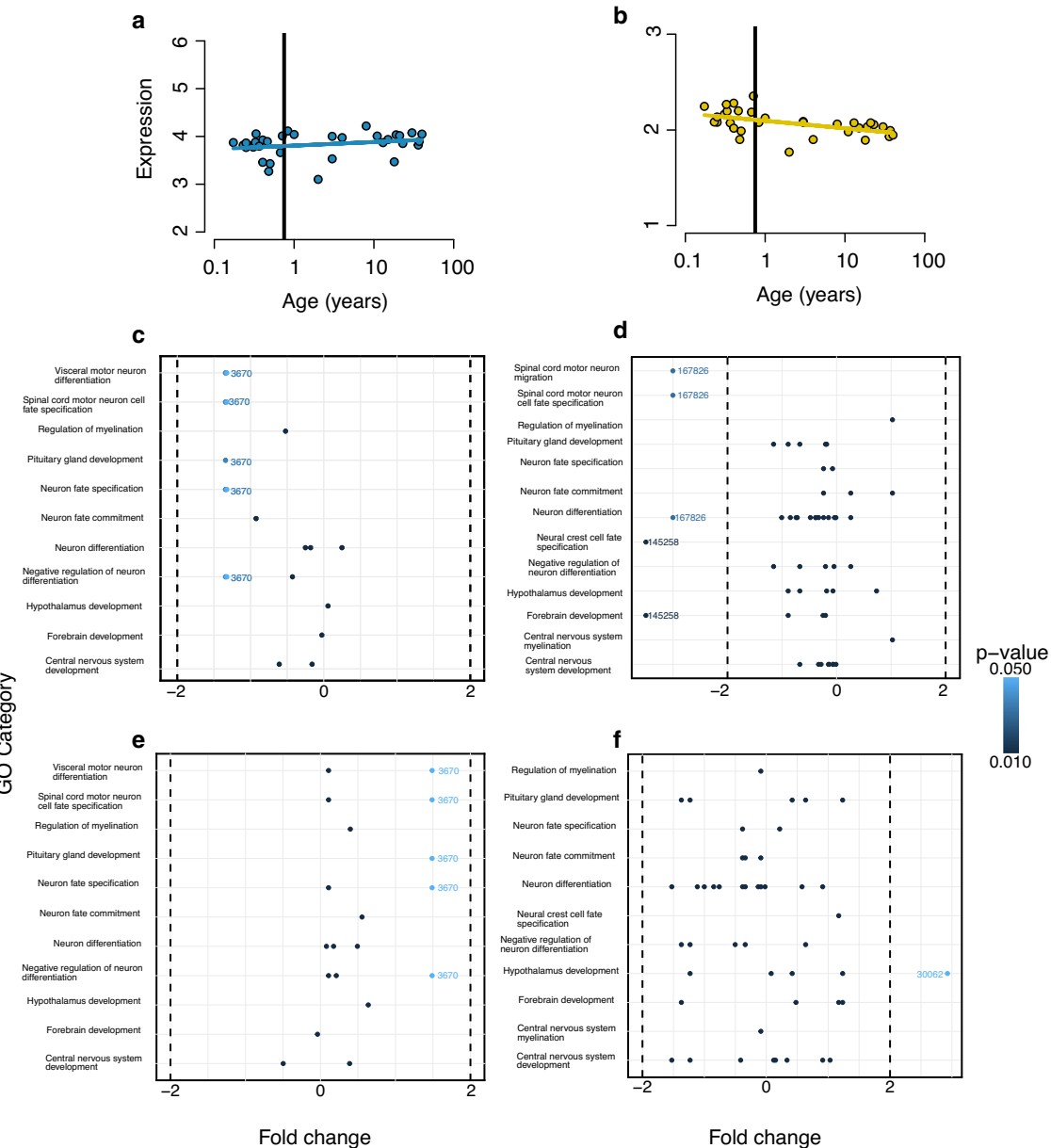

**Fig. 3 | Temporal patterns of cortical expression of SSD-associated genes and fold change analysis for sex-biased gene expression in the brain among SSD-associated genes.** Panels show average gene expression levels for (**a**) SSD-associated genes undergoing expansion on average for humans and (slope 0.08). **b** SSD-associated genes are undergoing contraction on average for humans (slope −0.08). The solid dark line represents when gestation time finishes (9 months). Gene expression data was log2($x$ + 1) transformed to account for zero expression values and mitigate the impact of low expression levels. Panels showing statistically significant female-to-male gene expression log2-fold change compared to GO categories in (**c**) prenatal brain for expanding SSD-associated genes, **d** prenatal brain for contracting SSD-associated genes, **e** adult brain for expanding SSD-associated genes and **f** adult brain for contracting SSD-associated genes. The dashed dark line marks the log2-fold change of 2 and −2, representing when fold change values are four times as large as those observed in the other phenotype. Source data are provided as a source Data File.

developmental stages. Moreover, our selection was supported by a study conducted by Carnoso-Moreira et al.[78], suggesting that human transcriptomics data offers valuable insights into shared cross-species evolutionary paths. Certain mammals exhibit a common evolutionary pattern in the temporal trajectories of brain gene expression. Our findings in humans reveal that contracting SSD-associated genes exhibit significantly higher expression in the brain during adulthood, consistent with their proposed role in brain development and functioning. However, this is only consistent across some life stages. Further studies exploring cross-species transcriptome profiles are needed to confirm and generalise our findings regarding the prominence of SSD-associated gene expression in the mammalian brain.

The enrichment analysis and transcriptional characterisation presented in this work provide evidence for an evolutionary link between sexual size dimorphism and brain evolution. In mammals, the evolution of SSD is mainly attributed to selective pressures acting on male body size in the context of sexual selection[3,79]. Polygynous populations, where larger males outcompete smaller ones in competition for access to multiple mates[80], exhibit higher rates of SSD. In contrast, monogamous species, often characterised by biparental care, typically display lower rates of SSD[43,81]. Such behavioural phenotypes accompanying monogamous mating systems likely result from changes in complex social skills leading to changes in brain function and size (see Schillaci 2006[41] for an example in primates). In birds and

mammals, species with pair-bonded mating systems present the largest brains[21,41,82], supporting a potential connection between brain evolution and sexual selection. However, natural and sexual selection are not necessarily mutually exclusive[45,48,83]. Ecological selection pressures may also significantly contribute to sexual size dimorphism[84] through competitive displacement, binomial and dimorphic niches[85]. Evidence suggests that in some clades, such as pinnipeds, sexual size dimorphism arose prior to the emergence of polygyny[86,87], showing the multifactor nature of the origin of complex phenotypes such as sexual size dimorphism.

This study reveals that within the SSD-associated genes expressed in the brain undergoing expansion and contraction, there are marginal differences in sex-biased gene expression, with three genes highly downregulated in females compared to males. Although thousands of genes have been identified as sex-biased in different species, and the high turnover in sex-biased genes makes it possible to create cross-species patterns, these assumptions should be made carefully, accounting for evolutionary history and genetic differences between species[24,88].

Our findings illustrate potential selection pressures affecting mammalian brain evolution in the context of SSD, often used as a proxy of sexual selection. This perspective potentially differs from Geoffrey Miller's "Mating Mind Hypothesis", which focuses on the evolution of human cognitive and behavioural traits through sexual selection[89]. However, our study exploring gene family evolution associated with SSD and brain development across 124 mammalian species, offers a broader perspective within the evolutionary framework. We suggest further investigation that could unveil the potential impact of sexual selection in the mammalian brain and behaviour evolution through more complex multilevel studies. Additionally, examining testis size with a comparative genomics scope may provide an integrative perspective on the molecular mechanisms underlying sex-specific selective processes affecting body size, sexual traits, and brain physiology. This research avenue is promising because investment in testis often trades-off against brain size due to the high metabolic costs of both tissues[20], brain and testis share several biochemical characteristics and may exhibit similarities in protein expression since both organs share similar gene expression patterns, at least in humans[90].

The evolution of the mammalian brain can follow different pathways as it directs an animal's interaction with its environment[91]. The gene family size changes observed in this work may be influenced by shared ecological pressures alongside sexual selection. For instance, the instrumental hypothesis states that species develop larger brains to forage more efficiently, while the social hypothesis suggests that large brains have evolved for the need to gain key social skills[92]. Investigating the interplay of the social taxa proportion, foraging strategies, and parental care in relation to sexual size dimorphism and gene family size changes could provide a better understanding of genome evolution dynamics.

In summary, our investigation unveils significant shifts in gene family size associated with SSD. The expanding SSD-associated gene families present enrichment in sensory perception of smell, while the contracting gene families display enrichment in various brain development and function roles. Reinforcing these findings, genes within these families exhibit pronounced expression in the adult brain compared to other tissues. These findings are consistent with selective pressures operating on brain development in monomorphic species, often characterised by monogamous mating systems, suggesting an imperative for complex social skills, parenting and highly developed brain functions in males and females. Our analyses withstand rigorous phylogenetic correction and are not explained by covariance between SSD and body mass. This study delves into the exploration of the genomic correlates across a major vertebrate clade with discernible association with SSD.

## Methods

### Male and female body mass data
Adult male and female body mass data were collected for 124 mammalian species from available literature and online sources, databases (e.g. The Animal Diversity Web[93]), literature and institutional datasets (Fig. 1 and Supplementary data 4). The average body mass (g) for the species was calculated by averaging adult male and female body mass per species. SSD was calculated as the log2-transformed ratio of average male versus average female body mass, as described by Dunham et al.[94]. We transformed (log10) male and female body mass to normalise the data.

### Test for Rensch's rule
To evaluate whether our data follows Rensch's rule, we conducted a major axis regression[95]. This analysis examines the relationship between two variables, regardless of which is taken as the dependent or independent variable. Major axis regression is distinguished by its adoption of a symmetrical nature, ensuring a balanced assessment of the relationship between variables. This method treats both variables impartially, accounting for uncertainties in both axes, enhancing statistical robustness[96]. Since Rensch's rule is indicated by an allometric slope between males and females exceeding 1, we tested whether the resulting slope differed from 1 (unity) using the R package "smatr"[97].

### Gene annotations and phylogenetic relationships
We obtained protein-coding gene annotations, corresponding coding sequences (CDS), and reference gene sequences (RefSeq) (Supplementary Data 5) for 146 species of mammalian species with a fully sequenced genome from the NCBI FTP site 16/08/20[98]. A phylogenetic tree containing 142 of the 146 of the above-described set of species was downloaded (16/08/20) from Timetree[99]; the not overlapping species between the phylogenetic tree and the list of species with an available genome were excluded from further analyses.

### Gene family annotation
Gene families for 142 species were annotated using Orthofinder[100]. Initially, we selected the longest available CDS sequence for each gene. Subsequently, all remaining CDS sequences within and between species were aligned with "DIAMOND"[101], as it is known for its speed, high sensitivity and scalability needed to handle large datasets[100]. Orthologue gene groups were constructed utilising a predefined phylogenetic tree, as described above. Orthofider's gene family estimation process involves, partitioning genes into groups predicted upon their evolutionary trajectory across a specified phylogeny. This ensures that genes with a common ancestor are placed together in the same group, called an orthogroup. Subsequently, Orthofinder constructs gene trees for each orthogroup, revealing the evolutionary dynamics within gene families.

### Gene family size analyses
To identify gene families associated with SSD to be included in the analysis, gene families were required to be present in at least 80% of species to filter out those that are lineage-specific and have at least three genes in at least one species to avoid gene presence-absence comparisons. Additionally, gene families with no variation in gene number across species were removed from the analysis, as regression analyses aim to construct models to explain significant amounts of variance. The same gene families were associated with SSD in an analysis that included all gene families (Supplementary Data 6; Supplementary Fig. 5).

Using the selected gene counts per gene family per species, we conducted a two-predictor PGLS[102,103] to examine associations between gene family size and the focal traits across 124 species (gene family size ~ SSD + log10-transformed average body mass) for which gene

family annotations and phenotype data were available. We employed the "nlme" v.3.1-152 R package[104], assuming a Brownian motion model of evolution. Post PGLS execution, we calculated $r$ values from $t$ values and adjusted $p$-values for multiple testing using the Benjamini–Hochberg correction.

Gene families presenting a significant positive correlation with gene family size ($r > 0.3$) were catalogued as expanding gene families. While contracting gene families were denominated when presenting a significant negative correlation with gene family size ($r < -0.3$).

Furthermore, we conducted a similar analysis to assess the relationship between gene family size and relative brain size (gene family size ~ SSD + Relative brain size), following the same methodology described above used for examining the association between gene family expansions and contractions with SSD and log10-transformed average body mass. This analysis was performed on a subset of 57 species for which both relative brain size and SSD data were available (Supplementary Data 7). Relative brain size was calculated using brain size controlled by the allometric effect of body mass by calculating the residuals of a log–log least-squares linear regression of brain size against body mass[105].

### Gene ontology term enrichment analysis

Functional term annotations from Gene Ontology Consortium database (GO)[106] were downloaded for each of the 124 mammalian species (NCBI FTP[98]). GO terms were linked to a gene family whenever that term was assigned to any gene within the gene family in any of the 124 species. For this section, we only used the gene families significantly associated with the gene family size analysis. We categorised GO terms annotated to fewer than 50 gene families, were pulled together into a category termed "small GO," and subsequently excluded from the analysis. This approach prioritised GO terms with more extensive association with gene families, thereby implying potential functional significance[33]. To assess the enrichment of GO categories among the gene families associated with the focal phenotype, we compared the proportion of gene families assigned to each GO term with the proportion of gene families assigned to the same GO term in 10,000 equally sized random samples from the background set. $Z$ scores were calculated from the mean and standard deviation for each GO term from the set of 10,000 randomised samples to determine the corresponding $p$-values adjusted using the Benjamini–Hochberg correction for multiple testing[40], as implemented in Castillo-Morales et al.[33].

### Gene expression in the human brain analysis

Genes were retrieved from the significantly associated gene families from the SSD + log10-transformed average body mass PGLS to assess expression patterns in the brain over time. Gene expression data for 18,947 protein-coding genes were downloaded from BrainSpan[107], with 933 of the genes overlapping with our SSD + log10-transformed average body mass PGLS dataset, covering multiple brain regions across 31 temporal intervals in human subjects. Average gene expression was calculated for three broad brain structures: cortex (Ocx, M1C-S1C, STC, MFC, DFC, OFC, ITC, HIP, VFC, PCx, TCx, A1C, V1C, M1C, IPC, S1C); subcortex (AMY, MGE, MD, CGE, DTH, STR) and cerebellum (URL, CB, CBC) (abbreviations explained in Supplementary Table 1). The resulting averages were used to calculate the overall brain average expression per gene. We conducted a log transformation of gene expression values using the formula $\log2(x + 1)$ to mitigate the impact of low or zero expression values. For samples taken at eight and nine weeks post-conception, the average expression per gene was calculated as the average of the samples available.

Then to conduct the gene expression ranking analysis in prenatal and adult stages, we used the SSD-associated genes from the gene families obtained in the SSD + log10-transformed average body mass PGLS. Human transcriptome data for 178 tissue samples were sourced from the Fantom database release 5[108]. Gene expression levels were averaged from the original 178 tissues for samples corresponding to brain areas and other tissues resulting in 49 adult and 20 prenatal healthy tissues, including the brain (Supplementary Data 2). For each gene, the number of tissues with gene expression levels higher than in the brain was calculated and averaged for SSD-associated genes. The statistical significance of gene expression in the brain was evaluated using a bootstrapping approach. We generated 10,000 bootstrap samples by randomly selecting the same number of genes from the dataset with replacements. Prenatal and adult tissues were processed separately.

Finally, to estimate sex-biased gene expression in humans, we used Brainspan data[107]. Fold change expression ratios were calculated as logarithm base 2 for female-to-male gene expression. These ratios were obtained for prenatal and adult stages for each gene. Later, $p$-values for each gene were calculated using one sample Wilcox test and then corrected by false discovery rate utilising the "stats" R package v. 4.2.3. Prenatal and adult stages were analysed separately.

### Reporting summary

Further information on research design is available in the Nature Portfolio Reporting Summary linked to this article.

## Data availability

All source data and datasets are provided with this paper. Supplementary Data 1 contains the gene family expansion/contraction analysis for sexual size dimorphism along with their corresponding statistical significance per gene family. Supplementary Data 2 provides the average gene expression levels in different tissues for adults and prenatal stages. Supplementary Data 3 includes the BrainSpan data (https://www.brainspan.org/static/download.html) used for the brain tissue-specific gene expression analysis. It includes Gene stable ID to Entrez Gene ID to GO category with gene expression and foldchange results with $p$-values. Supplementary Data 4 lists the species used in the analysis with phenotypical data and sources. Supplementary Data 5 presents the accession names of the CDS sequences for each species used for orthology mapping analysis, also including the download links from the Refseq FTP repository (https://ftp.ncbi.nlm.nih.gov/genomes/all/). Supplementary Data 6 contains the output from gene family expansion/contraction analysis, including zero variance gene families for sexual size dimorphism. Supplementary Data 7 lists the species with brain size used in the research, along with references. Lastly, Supplementary Table 1 provides a list of terms, abbreviations, and descriptions of brain structures used for sex-biased gene expression in temporal brain analysis. Supplementary files are available on the Figshare repository (https://doi.org/10.6084/m9.figshare.22770731). Source data are provided with this paper.

## Code availability

The code used for ortholog gene calculation, gene family size analysis, GO enrichment analysis and gene expression analyses can be found in the GitHub repository: https://github.com/animazum/SSD_genefamilysize.git.

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

## Acknowledgements

This study was supported by a University of Bath Ph.D. fee scholarship to B.P.-M.; TRNC Department of Higher Education and Foreign Affairs State Scholarship for Doctoral Studies, The Korner Travelling Fellowship Fund 2020 and Santander Mobility Award 2020 to H.K.; CONACyT PhD scholarships to A.P.A.-A. and A.C.-M.; a NERC GW4+ studentship (NE/L002434/1) to K.H.M., a travel grant SEP-UNAM-FUNAM Programa de Capacitación en Métodos de investigación, the Korner Travelling Fellowship Fund and TELMEX foundation fellowship to K.D.-B.; a Frontiers in Science CONACyT grant (No. FC-2016 /1682) and a Royal Society Newton Advanced Fellowship (no. NA160564) to T.S., D.C. and A.O.U.; a Frontiers in Science CONACyT (FC-2020/682142) to S.A., D.C., T.S. and A.O.U.; a NERC grant (NE/P004121/1) and PAPPIT-DGAPA-UNAM grant (IA204020) to A.O.U. We thank professors Laurence Hurst, Matthew Wills, Jason Wolf, and Dr. Nick Longrich for their valuable advice.

## Competing interests

The authors declare no competing interests.
