## [Peer Review File · Nature Communications]

Sexual size dimorphism in mammals is associated with changes in the size of gene families related to brain developmentREVIEWER COMMENTS

Reviewer #1 (Remarks to the Author):

In this manuscript, the authors present some very interesting and novel results – Gene family expansions are associated with sexual size dimorphism in mammals, and the SSD-associated gene families are more likely to exhibit sex-biased gene expression than non-associated gene families. However, there are many points in the manuscript that could use clarification (see specific comments), and I think the most interesting key results are not appropriately highlighted and not discussed fully.

Given that the focus of the paper is on gene family evolution, I think more of the introduction could be dedicated to helping the reader generate intuition for how gene family size relates to the evolution of novel traits. Some of the specific comments will help the authors identify particular places in the introduction where modifications might be appropriate, but I would also suggest condensing the information provided on SSD to more primarily focus on gene family size evolution.

In addition, all of the gene expression analysis in the paper is focused on the brains of humans, but the rationale for this is not stated. I suggest introducing and explaining the focus on brain tissues in the introduction so as not to take the reader by surprise.

Based on the title and abstract, the main takeaway seems to be that gene family expansions are associated with sexual size dimorphism, but this result does not appear in any of the figures. I think it would be useful to somehow represent these results in a figure (or a panel of figure 2, perhaps). For example, it would be useful to see the number of gene families with expansions (or contractions, if those were also found?) on the x-axis and SSD on the y-axis. Alternatively this could be added to the phylogeny in Fig 1 somehow (changing the colors of the branches to indicate the number of gene families that expanded perhaps?).

However, the results and discussion paint a slightly different picture: reversed SSD, which is not associated with sexual selection, is more strongly associated with gene family expansions than male-biased SSD. In addition, the figures do not very clearly portray whether gene families of interest expanded or contracted in association with greater SSD – and that most of the associations were negative. The discussion also does not emphasize these points, which seem important to the generalized understanding of how gene family size evolution might contribute to SSD evolution. Much of the discussion is focused on interpreting the GO associations, along with the importance of particular types of genes in the brain, which I do not think are the most exciting findings for a general audience. I suggest reducing the amount of space allocated to exploring the importance of GO associations, and rather focus on considering the evolutionary mechanisms and importance of the gene family expansions/contractions.

A final major concern is that the sources for the phenotypic data are not specified anywhere in the MS or the supplement, preventing the work from being fully reproducible.

Specific comments:

Lines 20-21: This sentence could make the directionality of the hypothesis clearer – are larger gene families reflective of more important pathways?

Line 44: The statement that male-biased SSD is associated with sexual selection strengths is supported by citations regarding only mammals and birds, and I think is not true for many invertebrates. Perhaps consider adding a caveat here ('in vertebrates' or perhaps 'in mammals and birds').

Lines 58-61: this final sentence felt repetitive of the previous paragraph.

Lines 66-67: the evidence for rapid evolution in sex-biased genes is not entirely clear; see Tosto et

al. (<https://doi.org/10.1038/s41559-023-02019-7>) for a recent review of the topic.

Line 73: I think there are some words missing from this sentence.

Lines 75-77: I would be interested to have the authors expand on this idea of mutation load a little bit more, specifically in the context of what that might look like or be detected in genome sequence data.

Lines 96-100: I do not think that the review of alignment tools is necessary for the introduction of this manuscript.

Lines 84-92: These examples are interesting but I think more specifics could be conveyed – specifically how do variations in gene family size provide insights into evolution of traits? Is it only expansion of gene families that suggests the importance of a gene family, or do contractions also indicate selection on the pathway? Are expanding gene families a source of novel gene functions (i.e., through duplications that are released from selection)?

Line 102: Did the authors hypothesize an increase or a decrease in gene family size associated with SSD? Or was it a two-tailed test initially? This should be more clearly stated here.

Line 110: Please specify how many mammalian species were studied.

Line 111: What are institutional databases? Are these publicly available? How might others access these datasets? I noted that Supplementary Table 1 does not include information on where the data were accessed from and all of this information should be provided to make the paper fully reproducible.

Line 119: The MAR acronym is only used a few times and only in this paragraph, so I suggest removing it for clarity.

Lines 118-119: This sentence repeats the previous sentence so can be removed.

Lines 121-122: This description of major axis regression was not entirely clear on why it is better to use than OLS. Consider rephrasing to emphasize both its symmetrical nature and its incorporation of error in the x variable and not just the y variable.

Line 128: Were these 146 species the same species used in the morphological dataset?

Lines 139-140: The total number of orthologs found is better suited for the results section.

Lines 145-146: I'm curious about why gene families without variation in number were removed – these are essentially ones with zero change, so removing these would bias your results towards finding a relationship between expanding gene family size and SSD. Wouldn't these be therefore be important to include? They essentially act as a control of sorts – if gene family size expansion is not in fact associated with SSD, these gene families would be equally likely to be associated with SSD.

Lines 165-166: I don't fully understand why GO terms with fewer than 50 gene families were pooled – can the authors please elaborate a bit on the rationale for this?

Lines 175-184: I'm confused by the purpose of the tissue expression analysis as described in this section. Was the goal to identify tissue specific patterns of expression, sex-biased patterns of expression, or something else? The aim should be included in the topic sentence of the paragraph (and also possibly in the sub-heading). I'm unclear on why both adults and prenatal tissues were used – I would expect massive differences in gene expression in these two groups due to their different developmental stages. I think this whole section needs to be re-written, but see the next few comments for clarity on specific points that were confusing.

Line 176: How representative of tissue specificity in mammals is human transcriptome data?

Earlier in the manuscript it was stated that there is rapid turnover of sex-biased gene expression patterns, but is this also true for tissue specificity?

Line 176: Were all of the 178 tissue samples brain samples (as implied in the next sentence)?

Lines 177-178: Why were brain data chosen as the focus of this analysis?

Line 181: 'Prominence of gene expression' is not a term I'm familiar with, please use more standard terminology. My guess is that it refers to identifying genes with the highest overall expression.

Lines 186-195: Why were brains the focus of this analysis? What is the relevance of brains to understanding SSD?

Lines 197-202: I suggest merging this section with the previous paragraph and re-naming that section something like: Identifying sex-biased gene expression using temporal brain transcriptome samples

Line 206: PGLS was already defined in the methods so you do not need to re-define the acronym.

Lines 245-246: The importance or significance of gene families remaining SSD associated after reversed SSD was removed was unclear – does this suggest that the relationship with SSD is strong or that it is weak? I think the authors could add a bit of interpretation to the results here.

Lines 251-268: Please include percentages of gene families that were associated with each of these categories to help readers understand the relative importance of each of the categories.

Line 262: I wouldn't have called an osteoblast an organ, consider rephrasing.

Lines 296-307: This analysis was not mentioned in the methods, I don't think, and its relevance was unclear and I'm not sure what it adds to the paper. If it remains in the manuscript, a description of the analysis should be included in the methods.

Line 310: The statement that SSD has been associated with sex differences in gene expression requires citations.

Lines 321-322: This first sentence could be more decisive – gene family expansion is associated with increased SSD? Decreased SSD? 'Varying levels of SSD' is not particularly informative.

Lines 325-327: So the results suggest that gene family expansion is not that important for male-biased SSD? Does this result suggest that sexual selection is not as important as other forms of selection on gene family expansion? If so, this seems counter to the main takeaways expressed in the title and the abstract.

Lines 330-334: This sentence is very long and quite confusing – I'm not sure what it is trying to say, please re-phrase.

Fig. 2: Please include the total number of gene families that are represented in each column, ideally on the graph itself (above or below each column) but at the very least in the figure caption.

Fig. 4: Are the SSD associated genes the ones that are in SSD-associated gene families, or were these genes identified through a different method?

Please improve the Fig. 4 and Supplemental Fig. 3 captions to make it clear the differences between them – I think that Fig. 4 is showing fold-change difference while the supplemental figure is overall expression, but the captions are not clear.

Lines 352-356: How might germ cell development be related to SSD, mechanistically?

Line 396: GFS is not an acronym that has been defined previously.

Lines 396-397: What evidence is there that gene family expansion is an indication of increased investment in that family's functions? Alternative explanations include relaxed selection on the family, by having fewer constraints on shifts in trait values due to new duplications? This is a point that I think needs to be more thoroughly explored in both the introduction and the discussion.

Line 407: 'both organs share an elevated number of genes' – please rephrase, as I believe this is meant to refer to gene expression (rather than total number of genes).

Lines 414-416: Please include citations for the studies that have examined transcriptome data from multiple species, and those that found sex-biased genes associated with brain function.

Lines 409-412: I suggest trying to summarize the key findings of your sex-biased gene analysis rather than re-stating what was done.

Lines 410-412: This work only conducted gene expression data between males and females in humans, as far as I understood, but this sentence implies that the results of this work are in comparison to previous work that only investigated sex-biased expression in one species. Please rewrite this sentence to more accurately reflect this research.

Lines 421-423: There appear to be some words and/or punctuation missing from this section.

Supplemental tables: Please make the README file more informative. Please ensure information on each file is included. See online resources such as:
<https://guides.lib.uci.edu/datamanagement/readme> for guides.

Reviewer #2 (Remarks to the Author):

Review of Sexual size dimorphism is associated with brain development gene family sizes in mammals by Padilla-Morales et al.

This article analyses the relationship of sexual size dimorphism (SDD) and gene family size. The authors perform a study of the size of the gene families in 124 mammalian genomes of SDD, reverse SDD or monomorphic species. They conclude that SSD is associated with family sizes of genes related with brain development. This analysis is therefore potentially interesting.

In my opinion, however, the MS is confusing and hard to follow in some parts, and the first part of the discussion is a repetition of the results or insubstantial lists of data with little biological significance. I believe that the text should be improved and extensively revised before it can be published.

There are many things that I do not understand:

1. The main message of the article is ambiguous. In the title, for instance, it is clearly stated that SSD is associated with family sizes of brain development genes, while in the abstract it is said that SSD-associated genes are enriched in germ cell development. It is confusing to me that that "SSD-associated gene families revealed a significant overrepresentation of functional categories related to germ cell development (lines 251-252)", but at the same time these "SSD-associated genes, on average brain expression had the highest rank (lines 281-282)", or with the fact that "SSD-associated genes had a lower expression during prenatal stages (line 291)", when they should be expressed if they are involved in germ cell development.

2. The title of the first section of the Results is "Gene family expansions associated with SSD and body mass (line 205)", but the authors demonstrate that "more gene families have expanded in line with the evolution of reversed SSD than with SSD (lines 328-329). In fact, the association between SSD and gene family sizes is more often negative than positive, 405 out of 754 gene

families (lines 212-219). Therefore, wouldn't it be more precise to entitle "Gene family reductions associated with SSD? The same applies for the Discussion, as authors states that "Our results provide evidence for size expansions of hundreds of gene families associated with varying levels of SSD across mammalian species (lines 321-322)".

3. The authors state that it has been shown that "as brain size increases throughout evolution, gene families have expanded" (lines 298-299). As far as I understand, this statement is contradictory to the authors' analysis since they found a negative association in most of their analyses: 27 out 52, or 239 out of 290 families negatively associated (lines 301 to 304).

4. SSD has commonly been associated with sex differences in gene expression, and therefore, the authors analyze whether SSD and reversed SSD-associated genes showed a higher degree of sex biased expression compared to background genes (lines 310-312). For this they calculated sex biased gene expression in the human brain and conclude that sex biased gene expression among SSD-associated genes was significantly higher than chance expectations (lines 314-315). Although this is true for the brain, does it respond to a general trait of SSD gene expression? What would happen if other organs or tissues were analyzed?

5. I understand that Figure 2A shows the functional annotation of the 340 gene families positively associated with SSD. Do all these families belong to a single GO group of germ cell development? I find it a bit strange.

6. The authors claim that reversed SSD-associated genes has a role in brain development, and because reversed SSD or monomorphic species tend to be monogamous, this association would support the social brain hypothesis. Which ones of the 124 analyzed species are monogamous and polygamous? Could this information be visualized in figure 2? Could the association between GFS and monogamy and polygamy be tested?

RESPONSE TO REVIEWERS' COMMENTS

We would like to express our sincere gratitude to the reviewers for dedicating their time and valuable expertise to evaluate our manuscript entitled "Sexual size dimorphism is associated with gene family contractions in brain development in mammals" submitted to Nature Communications. We greatly appreciate the constructive feedback provided, which has significantly enhanced the quality and clarity of our work.

In our subsequent response, we meticulously attend to each of the reviewers' comments. To facilitate easy reference, we have structured our replies in a point-by-point format corresponding to the comments. Additionally, we have supplied a Word document with track changes applied to the manuscript, clearly demarcating altered sections along with designated line numbers for seamless cross-referencing. In the majority of cases, we have integrated the revised segments into the manuscript to reflect the suggested changes.

We would like to mention that while we were addressing corrections, we spotted an error that only affects the GO enrichment analysis. The mistake was that gene families were duplicated, and categories were overrepresented (the new version of the script `GOenrichment.Analysis.R`, was corrected in the line 222). We fixed the error and thankfully over all the result was robust enough to sustain similar conclusions. While the associations with "synaptic plasticity" is no longer significant, we observe a significant association with "sensory perception of smell" in SSD-associated gene families under expansion. We deeply apologise of this error; we hope that you still find our research suitable for potential publication in Nature Communications. As a response to this issue, we have checked all scripts and analyses and we did not find any other mistake, we are sorry for this inconsistency.

We consider that the revisions made to the manuscript have substantially strengthened its scientific rigor and clarity. Therefore, we believe that the updated version is now well-aligned with the high standards of Nature Communications. Below, we outline our responses to the reviewers' comments.

We are very grateful for your feedback and comprehension.

- **Reviewer #1 (Remarks to the Author):**

In this manuscript, the authors present some very interesting and novel results – Gene family expansions are associated with sexual size dimorphism in mammals, and the SSD-associated gene families are more likely to exhibit sex-biased gene expression than non-associated gene families. However, there are many points in the manuscript that could use clarification (see specific comments), and I think the most interesting key results are not appropriately highlighted and not discussed fully.

Given that the focus of the paper is on gene family evolution, I think more of the introduction could be dedicated to helping the reader generate intuition for how gene family size relates to the evolution of novel traits. Some of the specific comments will help the authors identify particular places in the introduction where modifications might be appropriate, but I would also suggest condensing the information provided on SSD to more primarily focus on gene family size evolution. **R= Thank you for highlighting this, changes to the introduction were made to fit the recommendations.**

In addition, all of the gene expression analysis in the paper is focused on the brains of humans, but the rationale for this is not stated. I suggest introducing and explaining the focus on brain tissues in the introduction so as not to take the reader by surprise.

R= We thank you for this comment, We have addressed more the concept of brain evolution in the introduction to resolve this comment.

Based on the title and abstract, the main takeaway seems to be that gene family expansions are associated with sexual size dimorphism, but this result does not appear in any of the figures. I think it would be useful to somehow represent these results in a figure (or a panel of figure 2, perhaps). For example, it would be useful to see the number of gene families with expansions (or contractions, if those were also found?) on the x-axis and SSD on the y-axis. Alternatively this could be added to the phylogeny in Fig 1 somehow (changing the colors of the branches to indicate the number of gene families that expanded perhaps?). **R=Thank you for this comment, these changes have been addressed and how the figure 1 includes that data.**

However, the results and discussion paint a slightly different picture: reversed SSD, which is not associated with sexual selection, is more strongly associated with gene family expansions than male-biased SSD. In addition, the figures do not very clearly portray whether gene families of

interest expanded or contracted in association with greater SSD – and that most of the associations were negative. The discussion also does not emphasize these points, which seem important to the generalized understanding of how gene family size evolution might contribute to SSD evolution. Much of the discussion is focused on interpreting the GO associations, along with the importance of particular types of genes in the brain, which I do not think are the most exciting findings for a general audience. I suggest reducing the amount of space allocated to exploring the importance of GO associations, and rather focus on considering the evolutionary mechanisms and importance of the gene family expansions/contractions.

R= Thank you for pointing this out, we have addressed this in the figures, the narrative of the paper is focused on gene family expansion/contractions. Also, we have added more discussion about evolutionary mechanisms in conclusions (e.g. functional innovation, gene duplications and gene loss, response to selection pressures)

A final major concern is that the sources for the phenotypic data are not specified anywhere in the MS or the supplement, preventing the work from being fully reproducible. **R= I am sorry for this mistake, the sources are included in supplementary data 3**

Specific comments:

1- Lines 20-21: This sentence could make the directionality of the hypothesis clearer – are larger gene families reflective of more important pathways?

R= Thank you for this comment, we have rephrased the sentence to be more representative of the hypothesis that SSD is promoting changes in gene family size. “We would expect an association of sexual size dimorphism with evolutionary shifts in the function of molecular pathways, its connection to gene family expansion and contraction remains unexplored”. Lines 23 – 28

2- Line 44: The statement that male-biased SSD is associated with sexual selection strengths is supported by citations regarding only mammals and birds, and I think is not true for many invertebrates. Perhaps consider adding a caveat here (‘in vertebrates’ or perhaps ‘in mammals and birds’).

R= We have addressed that by adding “In mammals and birds” at the beginning of the paragraph. Line 47

3- Lines 58-61: this final sentence felt of the previous paragraph.

R=We agree with your comment. The sentence was removed from this paragraph. Line 81

4- Lines 66-67: the evidence for rapid evolution in sex-biased genes is not entirely clear; see Tosto et al. (<https://doi.org/10.1038/s41559-023-02019-7>) for a recent review of the topic.

R= Thank you very much for this, the sentence has been rephrased to provide more clarity and the Tosto et al reference was added. “However, while such studies hint at an evolutionary connection with sexual selection or other selection pressures, the full extent of this association remains elusive²³”.
Line 67 – 69

5- Line 73: I think there are some words missing from this sentence.

R= Rephrased. “Although the prevalence of comparative genome sequence studies and their relationship to sexual selection appears to be relatively limited”. **Lines 83 – 84**

6- Lines 75-77: I would be interested to have the authors expand on this idea of mutation load a little bit more, specifically in the context of what that might look like or be detected in genome sequence data.

R=To avoid confusion, it might be best to remove this sentence since our study is not directly related to it. Line 85

7- Lines 96-100: I do not think that the review of alignment tools is necessary for the introduction of this manuscript.

R= Agree, this will be removed. Line 111

8- Lines 84-92: These examples are interesting but I think more specifics could be conveyed – specifically how do variations in gene family size provide insights into evolution of traits? Is it only expansion of gene families that suggests the importance of a gene family, or do contractions also indicate selection on the pathway? Are expanding gene families a source of novel gene functions (i.e., through duplications that are released from selection)?

R= We thank you for this comment. In the introduction, we have included a new section that covers gene family expansion and contractions. “Gene duplication has long been recognised as a major source of functional innovation in genomes³⁸. For example, gene duplication of developmental-related genes is considered to have played a major role in the evolution of several vertebrate features. The most clear-cut example is the duplication of the Hox gene clusters³⁹. Groups of genes originating from an ancestral single-copy gene form a gene family which can expand through further gene duplications or contract through gene deletions. Gene family size is very dynamic over time⁴⁰ and these variations can provide insights into changes in the relative functional relevance of molecular functions and the molecular basis of complex phenotypes^{41–43}”. **Lines 87 – 95**

9- Line 102: Did the authors hypothesize an increase or a decrease in gene family size associated with SSD? Or was it a two-tailed test initially? This should be more clearly stated here.

R= Interesting question. We hypothesise a change in gene family size. Also, we cannot say that it is a two-tailed test as PGLS is a regression and traditionally they are not two-tailed. “The central aim of this study was to unravel the potential associations between gene family size and SSD while controlling for the influence of body mass. Our analysis encompassed functional annotations, gene expression profiles, and their intricate interplay with SSD. In doing so, we delved into the significance of gene expression and its implications for sexual dimorphism and the evolution of brain function. Through this multi-layered approach, we offer novel insights into the underpinning of SSD and its link to molecular evolution”. **Lines 113 – 119**

10- Line 110: Please specify how many mammalian species were studied.

R= We have added: “124 mammalian species”, “examine associations between gene family size and the focal traits across 124 species (gene family size and SSD) for which gene family annotations and phenotype data was available”. Lines 345, 388 – 389

11- Line 111: What are institutional databases? Are these publicly available? How might others access these datasets? I noted that Supplementary Table 1 does not include information on where the data were accessed from and all of this information should be provided to make the paper fully reproducible.

R= Sorry for the inconclusion, Sources were added to Supplementary data 3. Line 347

12- Line 119: The MAR acronym is only used a few times and only in this paragraph, so I suggest removing it for clarity.

R= Removed. Line 357

13- Lines 118-119: This sentence repeats the previous sentence so can be removed.

R= Removed. Line 354

14- Lines 121-122: This description of major axis regression was not entirely clear on why it is better to use than OLS. Consider rephrasing to emphasize both its symmetrical nature and its incorporation of error in the x variable and not just the y variable.

R= Thank you for this comment, this section was revised to suit the recommendation better.

“This analysis enables the examination of the relationship between two variables, regardless of which is taken as the dependent or independent variable. Major axis regression is distinguished by its adoption of a symmetrical nature, ensuring a balanced assessment of the relationship between variables and enabling a comprehensive analysis. This method treats both variables impartially, taking into account measurement uncertainties in both axes, which leads to a more robust analysis³¹”. **Lines 355 – 361**

15- Line 128: Were these 146 species the same species used in the morphological dataset?

R= Thank you for spotting this, we only found morphological data for 124 species. This is described later in the Gene family size expansion analyses section. “Subsequently, we conducted a PGLS^{93,94} to examine associations between gene family size and the focal traits across 124 species (gene family size and SSD) for which gene family annotations and phenotype data was available”. **Line 388**

16- Lines 139-140: The total number of orthologs found is better suited for the results section.

R= Removed. Line 376

17- Lines 145-146: I’m curious about why gene families without variation in number were removed – these are essentially ones with zero change, so removing these would bias your results towards finding a relationship between expanding gene family size and SSD. Wouldn’t these be therefore be important to include? They essentially act as a control of sorts – if gene family size expansion is not in fact associated with SSD, these gene families would be equally likely to be associated with SSD.

R= We thank the reviewer for this comment. We have conducted a supplementary analysis not excluding gene families with zero variance in gene number across species. The results remain the same with no changes in the set of gene families associated with SSD.

“Supplementary analysis where gene families with no variance are not excluded resulted in the same set of families found to be associated with SSD, ruling out any concerns of potential bias caused from the exclusion of no variance gene families (Supplementary data 5).” **Lines 383 – 386**

18- Lines 165-166: I don't fully understand why GO terms with fewer than 50 gene families were pooled – can the authors please elaborate a bit on the rationale for this?

R= We are thankful for this comment, the sentences were revised. This approach used in Castillo-Morales et al, intends to use GO terms that have a higher association with the function associated with GO term. “We categorized GO terms annotated to fewer than 50 gene families were pulled together into a category termed “small GO” and subsequently excluded from the analysis. This approach prioritised GO terms with more extensive association with gene families, thereby implying potential functional significance⁴¹.” **Lines 409 – 412**

19- Lines 175-184: I'm confused by the purpose of the tissue expression analysis as described in this section. Was the goal to identify tissue specific patterns of expression, sex-biased patterns of expression, or something else? The aim should be included in the topic sentence of the paragraph (and also possibly in the sub-heading). I'm unclear on why both adults and prenatal tissues were used – I would expect massive differences in gene expression in these two groups due to their different developmental stages. I think this whole section needs to be re-written, but see the next few comments for clarity on specific points that were confusing.

R= Thank you for this very good comment. To improve the clarity of this section we have rephrased and rewrote different parts. Our purpose is to assess if the genes associated with gene family expansion/contractions are significantly expressed in the brain (as in our GO enrichment analysis we found significant association with brain function and brain development). Then as we are analysing dimorphism, we find it important to assess the sex-biased expression. In figure 2 there is a high difference between gestation stages and adult gene expression for SSD genes under expansion and contraction. “SSD-associated genes were found to be enriched in brain development functions. If these genes do support brain development, we would expect them to have higher gene expression in brain compared to other tissues and within the brain to be more highly expressed in prenatal stages compared to adult stages.” **Lines 208-210, 288-298**

20- Line 176: How representative of tissue specificity in mammals is human transcriptome data? Earlier in the manuscript it was stated that there is rapid turnover of sex-biased gene expression patterns, but is this also true for tissue specificity?

R= Thank you very much this comment. We took the human as a representative mammalian species to explore the potential of SSD associated genes in contributing to brain development after functional enrichment analyses suggested such a role. Human data was used for this exploration because of the amount of transcriptomic resources available for the species. Confirming that the prominence of SSD associated genes in brain and foetal brain stages for SSD associated genes will be found across the mammalian phylogeny is beyond the scope of this study. “We explored gene activity profiles for SSD genes, taking human as a representative species as transcriptomic resources are readily available for a large number of tissues and developmental stages. Future studies examining transcriptome profiles from multiple species are required to confirm that SSD associated genes are indeed prominently expressed in the developing mammalian brain” Lines 280 - 284

21- Line 176: Were all of the 178 tissue samples brain samples (as implied in the next sentence)?

R= Thank you for spotting this. The data set included brain and other tissues. In the next sentence, we added a new supplementary table that includes all the tissues used for this analysis. “Human transcriptome data for 178 tissue samples were sourced from the Fantom database release 5 97. Gene expression levels were averaged for samples corresponding to brain areas and other tissues resulting in a total of 49 adult and 20 prenatal healthy tissues, including the brain (Supplementary data 8).” Lines 423 – 426

22- Lines 177-178: Why were brain data chosen as the focus of this analysis?

R= Thank you for this comment. As the GO enrichment analysis pointed out contracting gene families with genes associated with brain development. As sexual selection has been related with changes in behaviour and brain morphology we wanted to know if the genes associated with expanding and contracting gene families were linked to the brain (e.g. Griesser et al <https://doi.org/10.1073/pnas.2121467120>, Schillaci <https://doi.org/10.1371/journal.pone.0000062> and Fitzpatrick et al. <https://doi.org/10.1111/j.1420-9101.2012.02520.x>). “Sexual dimorphism in birds and mammals extends beyond physical characteristics, influencing brain structure, intricate social behaviours and mating systems¹⁴⁻¹⁶. A noteworthy inverse correlation between brain size and SSD has emerged in mammals¹⁷. While strong sexual selection driving SSD has been linked to diminution of brain development in pinnipeds, the underlying mechanism of this trade-off remains elusive¹⁰. For instance, the guppy exhibits rapid evolution in relative brain size driven by divergent selection in both sexes, resulting in larger-brained males associated with smaller offspring

and reduced gut size¹⁸. These instances underscore the profound role of sexual selection in shaping brain size and development. Despite these insights, the intricate interplay between sexual selection and brain evolution yields mixed outcomes, warranting further comprehensive research¹⁹.” **Lines 53 - 62**

23- Line 181: ‘Prominence of gene expression’ is not a term I’m familiar with, please use more standard terminology. My guess is that it refers to identifying genes with the highest overall expression.

R=Thank you for comment. The term was changed to “gene expression”. In this part, we are comparing the average of gene expression vs the average of random sampling. “Statistical significance of gene expression in the brain was assessed by comparing the already calculated averages to averages from 1,000 randomly selected samples of the same number of genes” Lines 428 – 429

24- Lines 186-195: Why were brains the focus of this analysis? What is the relevance of brains to understanding SSD?

R=We are grateful for this comment. We decided to use the brain as a main focus as in there is evidence that sexual selection is capable of shaping behaviour and potentially brain structures (e.g. Griesser et al <https://doi.org/10.1073/pnas.2121467120>, Schillaci <https://doi.org/10.1371/journal.pone.0000062> and Fitzpatrick et al. <https://doi.org/10.1111/j.1420-9101.2012.02520.x>). This is supported by our results from GO enrichment, which revealed significant enrichment in categories related to brain development. Moreover, in the introduction we have added references supporting that SSD could be affecting brain evolution. “Sexual dimorphism in birds and mammals extends beyond physical characteristics, influencing brain structure, intricate social behaviours and mating systems¹⁴⁻¹⁶. A noteworthy inverse correlation between brain size and SSD has emerged in mammals¹⁷. While strong sexual selection driving SSD has been linked to diminution of brain development in pinnipeds, the underlying mechanism of this trade-off remains elusive¹⁰. For instance, the guppy exhibits rapid evolution in relative brain size driven by divergent selection in both sexes, resulting in larger-brained males associated with smaller offspring and reduced gut size¹⁸. These instances underscore the profound role of sexual selection in shaping brain size and development. Despite these insights, the intricate interplay between sexual selection and brain evolution yields mixed outcomes, warranting further comprehensive research¹⁹.” **Lines 53 – 62**

25- Lines 197-202: I suggest merging this section with the previous paragraph and re-naming that section something like: Identifying sex-biased gene expression using temporal brain transcriptome samples

R= Thank for this comment, both sections were merged and the title was changed as well. “Using Brainspan data ⁹⁸, fold change expression ratios were calculated as logarithm base 2 for female-to-male gene expression. These ratios were obtained for prenatal and adult stages for each gene. The average fold change for SSD-associated genes was calculated and compared against the average fold change in 1,000 randomly selected gene samples. Prenatal and adult stages were analysed separately.” **Lines 444 – 448**

26- Line 206: PGLS was already defined in the methods so you do not need to re-define the acronym.

R= Thank you, this time we will keep it here as the methods section was moved to the end of the manuscript, this is the first time that it is introduced in the text. Line 123

27- Lines 245-246: The importance or significance of gene families remaining SSD associated after reversed SSD was removed was unclear – does this suggest that the relationship with SSD is strong or that it is weak? I think the authors could add a bit of interpretation to the results here.

R=Thank you for spotting this, we have included an interpretation of this analysis. “To investigate if SSD-associated gene family size is influenced by the absence of male size biases or specifically driven by female size biases, we excluded the 18 species where females are larger than males from the analysis. After this removal, 266 out of 313 gene families remained SSD-associated, and 352 out of 400 gene families remained under contraction with SSD. These results suggest that the removal of species with female size bias does not mask the association of gene family size with male size bias. Importantly, all of these gene families were consistent with the sets associated with SSD when species with female size bias were included (Supplementary data 1).” **Lines 159 – 166**

28- Lines 251-268: Please include percentages of gene families that were associated with each of these categories to help readers understand the relative importance of each of the categories.

R= Thank you this will improve the paper. This will make a more understandable section, the percentages have been added. “In the characterisation of the 313 SSD-associated expanded gene families (ExSSD), we observed that out of the 38 gene families significantly enriched (figure 1b), there was a significant overrepresentation of functional categories related to germ cell development and sensory receptor of smell. These categories accounted for 1.92% and 7.03% respectively, of the gene families significantly associated in the gene

family expansions/contractions analysis (Figure 1b). This enrichment patterns persisted in gene families associated with SSD in a two predictor PGLS model, which also included body mass (ExSSD + body mass). Here, we observed a similar overrepresentation in sensory receptor of smell, comprising 6.77% of the gene families significantly associated in the gene family expansion/contraction analysis (Figure 1b).

Among the 400 SSD-associated gene family contractions (CoSSD) calculated in the gene family expansion/contraction analysis, 171 gene families were significantly enriched (Figure 1b). Most of the enriched categories were associated with various aspects of development, including cell development (spinal cord development, skin development, muscle organ development, animal organ morphogenesis), general development (multicellular organism development) and brain development. Brain development-related processes represent 8.75% of the total contracting families, encompassing neuromuscular junction development, neuron migration, neuron differentiation, forebrain development (Figure 1b). When incorporating body mass into the PGLS model for contracted gene families (CoSSD + body mass), a smaller subset of the above categories remained enriched plus spinal cord development (Figure 1b).” **Lines 179 – 196**

29- Line 262: I wouldn't have called an osteoblast an organ, consider rephrasing.

R= Thank you for the comment. This section was changed, after solving the GO enrichment issue, thank you for the recommendation. “Among SSD-associated gene family contractions (CoSSD), most enriched categories were associated with various aspects of development, cell development (spinal cord development, skin development, muscle organ development, animal organ morphogenesis) and general development (multicellular organism development). Among the gene families significantly associated in the gene family expansions/contractions analysis, brain development-related processes represent 8.75% of the total contracting families, encompassing neuromuscular junction development, neuron migration, neuron differentiation, forebrain development (Figure 1b). When incorporating body mass into the PGLS model for contracted gene families (CoSSD + body mass), a smaller subset of the above categories remained enriched plus spinal cord development (Figure 1b)” **Lines 187 – 196**

30- Lines 296-307: This analysis was not mentioned in the methods, I don't think, and its relevance was unclear and I'm not sure what it adds to the paper. If it remains in the manuscript, a description of the analysis should be included in the methods.

R= Thank you for highlighting this issue. As one of our main results is the contracting gene families associated with size dimorphism and several of those genes are linked with brain development function. We were interested in investigating the possible connection between the encephalization index and sexual size dimorphism as well. Therefore, we opted to run this analysis. This section is now described in the methods section. “Furthermore, we conducted a similar analysis to assess the relationship between gene family expansion and encephalization index, following the same methodology used for examining the association between gene family expansion and log average body mass in SSD. This analysis was performed on a subset of 57 species for which both encephalization index and SSD data were available (Supplementary

data 7). Encephalization index was calculated by estimating brain mass relative to body size, while accounting for the allometric effect, through the calculation of residuals from a log-log least squares linear regression of brain mass against body mass⁹⁵” **Lines 396 – 403**

31- Line 310: The statement that SSD has been associated with sex differences in gene expression requires citations.

R= Thank you for your comment. I will increase clarity in the text. References added. “Finally, as SSD has commonly been associated with sex differences in gene expression^{2,22,55}” **Line 224**

32- Lines 321-322: This first sentence could be more decisive – gene family expansion is associated with increased SSD? Decreased SSD? ‘Varying levels of SSD’ is not particularly informative.

R= The sentence has been rephrased, thank you for this contribution. “Our results provide compelling evidence of significant size expansions and contractions of hundreds of gene families associated with SSD across 124 mammalian species.” **Lines 235 – 236**

33- Lines 325-327: So the results suggest that gene family expansion is not that important for male-biased SSD? Does this result suggest that sexual selection is not as important as other forms of selection on gene family expansion? If so, this seems counter to the main takeaways expressed in the title and the abstract.

R= Sorry for the confusion, the text has been revised to ensure accuracy. Our results suggest that gene family size expansions associated with SSD is significantly important for genes with olfactory receptor and germ cell developmental function. Those two functions and SSD have a link with sexual selection. Meanwhile, gene families under contraction are present in species with lower SSD, but linked with brain developmental functions. This is also linked with sexual selection. “Above enrichment analysis and transcriptional characterization suggest an evolutionary link between sexual selection and brain evolution. In mammals, the evolution of male biased SSD is mainly attributed to selective pressures acting on male body size related to sexual selection^{3,73}; in polygynous populations, larger males outcompete smaller ones in competition for access to multiple mates⁷⁴. In contrast to polygynous species, monogamous species, often characterised by biparental care, tend to have lower rates of SSD^{26,75}. Such behavioural phenotypes that accompany monogamous mating systems likely result from changes in complex social skills leading to changes in brain function and size (see Schillaci 2006 for an example in primates). In birds and mammals, species with pair bonded mating systems have the largest brains^{14,15,76}, further supporting a link between sexual selection and brain evolution.” **Lines 284 – 294**

34- Lines 330-334: This sentence is very long and quite confusing – I’m not sure what it is trying to say, please re-phrase.

R= We are sorry for the confusing sentence. This part was removed from the conclusion section as the narrative was changed to address gene families under expansion or contraction linked with SSD rather than gene families associated with female biased SSD or male biased SSD. Lines 240

35- Fig. 2: Please include the total number of gene families that are represented in each column, ideally on the graph itself (above or below each column) but at the very least in the figure caption.

R= The number is now indicated in the figure 1, this suggestion makes the figure more informative. Line 469

36- Fig. 4: Are the SSD associated genes the ones that are in SSD-associated gene families, or were these genes identified through a different method?

R= Thank you for this contribution. Yes, the SSD-associated genes are the ones coming from the SSD-associated gene families. It is now stated in the results and methods sections more clearly that the genes in question come from the SSD-associated gene families, thank you for this recommendation. “In the next sections genes within SSD-associated gene families will be addressed as SSD-associated genes.”, “GO terms were linked to a gene family whenever that term was assigned to any gene within the gene family in any of the 124 species. For this section we only used the gene families significantly associated coming from the gene family size analysis” **Lines 176 – 177, 408 – 411**

37- Please improve the Fig. 4 and Supplemental Fig. 3 captions to make it clear the differences between them – I think that Fig. 4 is showing fold-change difference while the supplemental figure is overall expression, but the captions are not clear.

R= To provide clarity, the names have been modified, thank you for spotting this. Lines Figure 2. Lines 481,

supplementary figure 3. Line 513

38- Lines 352-356: How might germ cell development be related to SSD, mechanistically?

R= This comment will make text better, we have added that the genes found associate in the analysis are linked to sex differentiation, gamete differentiation plus other functions. “Gene families that have significantly expanded in line with SSD are enriched in functional terms associated with germ cell development. Genes within gene families with this functional term annotation (ENTREZ ID: 4090, 5914, 63946, 79727) are involved in inhibition of hematopoietic progenitor cells proliferation, transcription regulation of clock genes, male gamete differentiation, sex differentiation and posttranscriptional regulation of developmental genes^{61–64}.” **Lines 258 – 263**

39- Line 396: GFS is not an acronym that has been defined previously.

R= Agree with you, deleted. Line 331

40- Lines 396-397: What evidence is there that gene family expansion is an indication of increased investment in that family’s functions? Alternative explanations include relaxed selection on the family, by having fewer constraints on shifts in trait values due to new duplications? This is a point that I think needs to be more thoroughly explored in both the introduction and the discussion.

R=We have added references talking about gene duplication in more extent, this should increase the informative value of the manuscript. “Gene duplication has long been recognised as a major source of functional innovation in genomes³⁸. For example, gene duplication of developmental-related genes is considered to have played a major role in the evolution of several vertebrate features. The most clear-cut example is the duplication of the Hox gene clusters³⁹. Groups of genes originating from an ancestral single-copy gene form a gene family which can expand through further gene duplications or contract through gene deletions. Gene family size is very dynamic over time⁴⁰ and these variations can provide insights into changes in the relative functional relevance of molecular functions and the molecular basis of complex phenotypes^{41–43}.”, “Expansion and contractions of gene families hold important roles in shaping biological functions^{45,46}. In mammals, gene families

present a paradigm, exhibiting gains through gene duplication and losses via pseudogenization, thereby driving the adaptation to new environments^{45,47}. Gene duplication events are potent drivers of evolutionary processes^{48,49}, significantly influencing the early evolution of human olfactory receptor genes⁵⁰. These duplications are also present in placental mammals, where they correlate with olfactory capacity⁴⁷ and adaptation to varied ecological niches⁵¹. **Lines 87 – 95, 100 – 106.**

41- Line 407: 'both organs share an elevated number of genes' – please rephrase, as I believe this is meant to refer to gene expression (rather than total number of genes).

R=The sentence has been changed, sorry for the confusion. "This research avenue is promising because investment in testes often trades-off against brain size due to the high metabolic costs of both tissues⁷⁷, brain and testes share several biochemical characteristics and may exhibit similarities in protein expression, since both organs share similar **gene expression patterns**, at least in humans⁷⁸" **Line 304**

42- Lines 414-416: Please include citations for the studies that have examined transcriptome data from multiple species, and those that found sex-biased genes associated with brain function.

R= The citations were included, thank you for the recommendation. "Only a handful of studies have examined transcriptome data from multiple species at a time^{41,42}. Interestingly, these studies have found significant roles associated with brain function among sex-biased genes^{79,80}. " **Lines 310 – 312**

43- Lines 409-412: I suggest trying to summarize the key findings of your sex-biased gene analysis rather than re-stating what was done.

R= The sentence was rephrased to show the main finding of that section, thank you for your help in turning the sentence sharper. "This study indicated that within the SSD-associated genes that are undergoing expansion, a significant difference in the expression patterns between human males and females. However, this distinction is not evident among the genes undergoing contraction." **Lines 306 – 308**

44- Lines 410-412: This work only conducted gene expression data between males and females in humans, as far as I understood, but this sentence implies that the results of this work are in comparison to previous work that only investigated sex-biased expression in one species. Please rewrite this sentence to more accurately reflect this research.

R= The sentence was rewritten to be clearer, thank you. “This study indicated that within the SSD-associated genes that are undergoing expansion, a significant difference in the expression patterns between **human males and females**. However, this distinction is not evident among the genes undergoing contraction.”

Lines 307

44- Lines 421-423: There appear to be some words and/or punctuation missing from this section.

R= The section has been rewritten to be more precise, thank you. “In a recent study conducted by Tripp et al.⁸¹, the gene expression underlying the formation of pair-bonds was investigated in prairie voles, known for their tendency to form long-lasting, socially monogamous relationships, and in meadow voles, a closely related species known for their polygamous behavior⁸¹.” **Lines 315 – 319**

45- Supplemental tables: Please make the README file more informative. Please ensure information on each file is included. See online resources such as: <https://guides.lib.uci.edu/datamanagement/readme> for guides.

R= Readme file was changed to be more informative of the contents of each directory, thank you for this comment, thank you for the material.

Supplementary Data for the Paper "Sexual Size Dimorphism is Associated with Gene Family Contractions in Brain Development in Mammals"

Author: Benjamin Padilla-Morales

Affiliation: University of Bath

Email: benjaminpadillams@gmail.com

Overview

This repository contains supplementary tables for the article "Sexual size dimorphism shapes gene family contractions in brain development in Mammals." The study investigates gene family size expansions and contractions in relation to brain development and sexual size dimorphism in mammals, employing a comparative genomics approach. Subsequently, we perform a Gene Ontology (GO) enrichment analysis on the gene families experiencing expansion and contraction to uncover the functional significance of associated genes. Additionally, we conduct a gene expression analysis of genes linked to these families to determine their over expression in brain tissue, showing the potential influence of sexual selection on the mammalian brain.

Contents

- **Supplementary Figures:** Supplementary figures 1 to 4 with corresponding descriptions.
- **Supplementary data 1:** Output from gene family expansion/contraction analysis for sexual size dimorphism, along with their corresponding statistical significance per gene family.
- **Supplementary data 2:** List of Gene Ontology (GO) categories associated with body mass gene family expansion/contractions.
- **Supplementary data 3:** List of species used in the analysis with phenotypical data, including sources.
- **Supplementary data 4:** List of sequences with respective species used for the orthology mapping analysis.
- **Supplementary data 5:** Output from gene family expansion/contraction analysis including zero variance gene families for sexual size dimorphism.
- **Supplementary data 6:** Table with the 5425 gene family counts that passed the filtering process and were used in the gene family size expansion/contraction analysis.
- **Supplementary data 7:** List of species with encephalization index used in this research, along with references.
- **Supplementary data 8:** Average gene expression levels of brain areas and other tissues for adults and fetuses.
- **Supplementary table 1:** List of terms, abbreviations, and descriptions of brain structures used for sex-biased gene expression in temporal brain analysis.

For any questions or inquiries, please contact Benjamin Padilla-Morales at the provided email address.

- **Reviewer #2 (Remarks to the Author):**

Review of Sexual size dimorphism is associated with brain development gene family sizes in mammals by Padilla-Morales et al.

This article analyses the relationship of sexual size dimorphism (SDD) and gene family size. The authors perform a study of the size of the gene families in 124 mammalian genomes of SDD, reverse SDD or monomorphic species. They conclude that SSD is associated with family sizes of genes related with brain development. This analysis is therefore potentially interesting.

In my opinion, however, the MS is confusing and hard to follow in some parts, and the first part of the discussion is a repetition of the results or insubstantial lists of data with little biological significance. I believe that the text should be improved and extensively revised before it can be published.

There are many things that I do not understand:

1. The main message of the article is ambiguous. In the title, for instance, it is clearly stated that SSD is associated with family sizes of brain development genes, while in the abstract it is said that SSD-associated genes are enriched in germ cell development. It is confusing to me that that “SSD-associated gene families revealed a significant overrepresentation of functional categories related to germ cell development (lines 251-252)”, but at the same time these “SSD-associated genes, on average brain expression had the highest rank (lines 281-282)”, or with the fact that “SSD-associated genes had a lower expression during prenatal stages (line 291)”, when they should be express if they are involved in germ cell development.

R2= Thank you for the comment. To make things clearer, we have taken significant measures to address these aspects in the text, ensuring a more coherent message. Moreover, it is crucial to emphasize that SSD-associated genes with in gene families under expansion, exhibit connections not only with germ cell development but also with other functions, such as sensory perception of smell. By broadening the scope of gene functions associated with SSD, our findings offer a comprehensive understanding of the complex molecular mechanisms underlying sexual size dimorphism, and therefore sexual selection. Showing a high expression in the brain and lower expression in developmental stages.

2. The title of the first section of the Results is “Gene family expansions associated with SSD and body mass (line 205)”, but the authors demonstrate that “more gene families have expanded in line with the evolution of reversed SSD than with SSD (lines 328-329). In fact, the association between SSD and gene family sizes is more often negative than positive, 405 out of 754 gene families (lines 212-219). Therefore, wouldn't it be more precise to entitle “Gene family reductions associated with SSD? The same applies for the Discussion, as authors states that “Our results provide evidence for size expansions of hundreds of gene families associated with varying levels of SSD across mammalian species (lines 321-322)”.

R2= Very good recommendation, The title was changed to be more accurate “Sexual size dimorphism is associated with gene family contractions in brain development in mammals”. The statement in the conclusions section was changed to: “Our results provide compelling

evidence of significant size expansions and contractions of hundreds of gene families associated with SSD across 124 mammalian species, predominantly under contraction” Lines 236 – 237. I am sorry for the confusion, stating that gene family expansion and contraction associated with SSD is more understandable than stating gene family expansions associated with high or low levels of SSD. The narrative of the paper was changed to address expansions/contractions of gene families and therefore the genes within these families.

3. The authors state that it has been shown that "as brain size increases throughout evolution, gene families have expanded” (lines 298-299). As far as I understand, this statement is contradictory to the authors’ analysis since they found a negative association in most of their analyses: 27 out 52, or 239 out of 290 families negatively associated (lines 301 to 304).

R2= I am sorry for the confusion of this analysis, the PGLS analysis reveals that gene families demonstrate different patterns based on their association with Sexual Size Dimorphism (SSD). For instance, positive associations with SSD indicate that these gene families tend to expand among species with increasing SSD. Conversely, negative associations with SSD imply that gene families are more likely to expand in species with lower SSD. Nevertheless, I agree with changing the narrative to make it more understandable by stating that positively associated gene families are expanding and negatively correlated are contracting. This change is not modifying the interpretation, it is just showing the results in a more dynamic and less complex way.

4. SSD has commonly been associated with sex differences in gene expression, and therefore, the authors analyze whether SSD and reversed SSD-associated genes showed a higher degree of sex biased expression compared to background genes (lines 310-312). For this they calculated sex biased gene expression in the human brain and conclude that sex biased gene expression among SSD-associated genes was significantly higher than chance expectations (lines 314-315). Although this is true for the brain, does it respond to a general trait of SSD gene expression? What would happen if other organs or tissues were analyzed?

R2= Thank you for highlighting this point. In the “Expression patterns of SSD-associated genes in brain tissue” section of our manuscript, we examined the expression of the significantly associated genes in various human tissues (49 tissues) and found that some of them showed significant expression in other tissues, but in most cases the brain was the highest-ranking tissue. Also, the manuscript is focused to analyse sexual selection and its link with the brain, therefore exploring beyond that would be out of the scope of the paper for the sex-biased gene expression analyses.

5. I understand that Figure 2A shows the functional annotation of the 340 gene families positively associated with SSD. Do all these families belong to a single GO group of germ cell development? I find it a bit strange.

R2= Thank you for highlighting this potential misconception. From all of the significantly associated gene families under expansion with SSD (313 gene families) in the gene family size analysis, 1.92% gene families showed significant enrichment in the Gene Ontology analysis for germ cell development. Also, we have 38 gene families in total that appear are enriched for significantly associated gene families under expansion with SSD. Also, we need to keep in mind that GO categories are limited, and we could be having a loss of information because we do not have access to more than the available GO categories already published. We addressed this comment in the text more clearly “In the characterisation of the 313 SSD-associated expanded gene families (ExSSD), we observed that out of the 38 gene families significantly enriched (figure 1b), there was a significant overrepresentation of functional categories related to **germ cell development and sensory receptor of smell. These categories accounted for 1.92% and 7.03% respectively**, of the gene families significantly associated in the gene family expansions/contractions analysis (Figure 1b). This enrichment patterns persisted in gene families associated with SSD in a two predictor PGLS model, which also included body mass (ExSSD + body mass). **Here, we observed a similar overrepresentation in sensory receptor of smell, comprising 6.77% of the gene families significantly associated in the gene family expansion/contraction analysis** (Figure 1b).

Among the 400 SSD-associated gene family contractions (CoSSD) calculated in the gene family expansion/contraction analysis, 171 gene families were significantly enriched (Figure 1b). Most of the enriched categories were associated with various aspects of development, including cell development (spinal cord development, skin development, muscle organ development, animal organ morphogenesis), general development (multicellular organism development) and brain development. **Brain development-related processes represent 8.75% of the total contracting families, encompassing neuromuscular junction development, neuron migration, neuron differentiation, forebrain development** (Figure 1b). When incorporating body mass into the PGLS model for contracted gene families (CoSSD + body mass), a smaller subset of the above categories remained enriched plus spinal cord development (Figure 1b).” **Lines 180 -199**

6. The authors claim that reversed SSD-associated genes has a role in brain development, and because reversed SSD or monomorphic species tend to be monogamous, this association would support the social brain hypothesis. Which ones of the 124 analyzed species are monogamous and polygamous? Could this information be visualized in figure 2? Could the association between GFS and monogamy and polygamy be tested?

R2= Very good comment, we were intrigued by this idea and explored it prior to submission, gathering data for approximately 80 species. Regrettably, our analysis did not reveal a significant correlation between the trait and gene family size. Consequently, we decided to

exclude this particular result from the manuscript, as its inclusion might introduce unnecessary confusion.

REVIEWER COMMENTS

Reviewer #1 (Remarks to the Author):

Major comments

The authors have clarified the focus of the manuscript and addressed many of my comments from the first round of review. I felt generally that the overall focus is much improved, although I have a number of suggestions for further improvement of the text – mainly, when topic sentences do not match the actual topics of the paragraphs. The improved flow of the manuscript led me to identify several concerns about the statistical analyses:

(Stats comment 1) The authors responded to my comment regarding removing gene families that did not expand or contract by saying that it did not change the results – this is a positive sign! However, the data file they shared would require a fair amount of work to interpret, so it is difficult for me to evaluate their assertion. I suggest including a supplementary figure – or, if it really makes no difference, include the dataset with all of the gene families included.

(Stats comment 2) The authors present the results of many phylogenetics least squares corrected regressions, including those of a model including just SSD, just body mass, and the joint effects of SSD and body mass. I am concerned by this – if the joint effects of the two factors are of interest, the model including both (and their interaction) should be presented and interpreted, not all three separate models. Analysing all three models could inflate type I error rates and seems somewhat misleading. I suggest the authors focus on the joint model and interpret the results from that model.

(Stats comment 3) Gene expression profiles – comparing to 1000 random genes is not a particularly robust approach. Consider using proper bootstrapping/jackknifing analyses or implementing a statistical comparison of expression between tissues/ages in a package like DESeq2 or edgeR.

In addition, the authors assume in a number of places that selection is the driving force behind gene family expansion or contraction, but one could expect some degree of gene family size change due to mutation and drift. The manuscript would benefit from ensuring that statements are not over-reaching, and that statements about selection (when not actually tested/modelled) are phrased as hypotheses for explaining the patterns arising from the data.

When I first reviewed the manuscript, I had not realised the number of contracting gene families is actually larger than the number of gene family expansions. Do the authors have any intuition for whether contracting gene families are loss of whole segments of DNA or is it due to pseudogenisation of genes? This would be a useful addition to the discussion, in my opinion.

Specific comments

Abstract: Perhaps mention that more contractions occurred than expansions, potentially.

Lines 31-32: The authors make the assumption that contracting gene families are due to selective pressures, but isn't it equally possible that gene family contractions are due to a relaxation of selection pressures, and loss of genes due to mutation and drift?

Lines 53-62: The opening sentence of this paragraph focusses on mammals and birds, but then the focal example is in guppies (neither a mammal nor a bird). Consider re-framing this paragraph.

Lines 70-72: Check this sentence – I believe there is unnecessary repetition of a phrase.

Lines 75-84: From this paragraph, I remain unclear on how comparative genomics offers insight

into Rensch's rule. Some of the information about sexual size dimorphism in this paragraph is also somewhat included in the first few paragraphs, so the authors might choose to delete this paragraph entirely, or merge it with for example paragraph 2 of the introduction.

Line 101: Consider rephrasing "gene families present a paradigm" as it was unclear what was meant by this statement.

Lines 105-106: In the previous work, did more duplications result in expanded olfactory capacity and ability to adapt to an increased variety of niches? Or was it a negative correlation and smaller gene families was associated with broader sets of traits? A clearer description of previous results related to gene family expansions/contractions will help the reader set expectations for your study.

Line 108: Please specify what is being compared when mentioning 'differences and similarities'

Lines 109-111: This sentence is sufficiently vague that I do not know what the authors are encouraging the reader to consider.

Lines 115-116: The analysis of gene expression profiles was limited to humans, so was not really intricately interplaying with SSD in this paper. Consider re-phrasing to better reflect what was actually done in this manuscript.

Line 134: Change negatively to negative.

Line 135: Change less to fewer.

Lines 140-145: I suggest stating that body mass was log₁₀-transformed here, given that the results now come before the methods.

Lines 123-125: Consider more clearly stating how many gene families total were included in the analysis – I believe it was 5425, based on the n reported on line 125. Consider also re-iterating that this analysis included 124 species.

Lines 149-156: If the dual effects of SSD and body mass were of interest to the authors from the start, it seems unnecessary (and possibly inappropriate) to interpret results of both models that did not include both terms.

Lines 167-177: The encephalization index comes out of nowhere – encephalization should probably be introduced as a relevant topic in the introduction.

Lines 168-169: This previous investigation into gene family expansions and brain size should definitely be featured in the introduction – this would be more relevant than the guppy example, probably.

Lines 197-198: I'm unclear on which analysis this sentence is referring to – is this the same model that was discussed on lines 138-148? Wouldn't this model have included all gene families, including those that both expanded and contracted? I think I am getting confused by the way the analyses are referred to in parentheses, but I think the authors need to clarify this for the results to be interpretable.

Lines 210-211: Please articulate the number of samples from each tissue type at adult vs prenatal stages.

Line 214: How was this random expectation of brain expression defined? Wouldn't you rather know whether these genes were up-regulated in adult brains relative to other adult tissues? And for contracting genes, what are 'randomly selected gene samples'? See my major comment re: statistical analysis.

Lines 219-235: It would be good to clarify in each of these sub-sections that these analyses are referring to humans only. I also encourage the authors to consider whether making inferences

about sex biased expression patterns based only on human data is relevant to gene families across mammals, especially given that genes have generally high turnover in their sex bias (e.g., <https://doi.org/10.1126/science.aaw7317>, <https://doi.org/10.1126/science.aaw7317>). Are there any gene families that expanded/contracted in humans (and/or primates) that could be a better focus for these interpretations?

Lines 244-246: The sentence about the associations being associated was very confusing, please rephrase.

Lines 260-268: The GO associations for gene family expansions being related to gametes seems related to the similar research about mammalian testes size being related to the strength of postcopulatory sexual selection. I think the authors mention this somewhere else, but it seems as though it would be a relevant focus for this paragraph. Consider splitting the discussion of expanding gene family associations into its own paragraph from the contracting gene families.

Lines 308-310: If gene families are contracting, I would think the members are less likely to have sex- and tissue-specific expression patterns – they will have more constraints/shared functions. Therefore I think this observation is not unexpected. I think this section could be improved by focussing on mechanisms and/or thinking about these patterns from this slightly more gene-centric view.

Lines 312-313: More than references 41 and 42 have studied transcriptome data from multiple species. For example, see the following (and there are probably more):

<https://doi.org/10.1111/mec.15115>
<https://doi.org/10.1126/science.aaw7317>
<https://doi.org/10.1126/science.aaw7317>
<https://doi.org/10.1126/science.adf1046>
<https://doi.org/10.1111/evo.14579>
<https://doi.org/10.1111/mec.13596>
<https://doi.org/10.1073/pnas.1501339112>

Lines 313-314: Re-emphasising the brain results here seems out of place – consider moving to the other sections where the brain results are discussed.

Lines 315-326: The link from the results in this paper to this example by Tripp et al was not very clear and I felt detracted from the key messages of the manuscript.

Lines 371-373: I thought that only 124 species were retained because that was the number that had available body size data. Please clarify the number of species used in downstream analyses.

Lines 385-386: I suggest rephrasing to “The same gene families were associated with SSD in an analysis where all gene families were included (Supplementary data 5)”. Although, this would be better represented as a supplementary figure (perhaps instead of the data file, or in addition to) – the file is difficult to interpret and compare easily to the main results presented, so it is difficult for me to evaluate whether my concern is addressed.

Lines 400-401: I’m confused about which analysis the methodology was based off of, and whether encephalization and SSD were included in this analysis.

Lines 405-406: I am confused about which variables went into this model, please rephrase. Was body size or SSD included as explanatory/random variables, or was it just the encephalization index?

Line 427: Here in the methods it states that 178 tissue samples were sourced, but the methods describes using 49 adult and 20 prenatal tissues. Please clarify in the main text.

Lines 430-432: Why was gene expression in the central nervous system used as the point of

comparison for the brain expression?

Lines 433-435: Comparing the expression to 1000 random samples seems a bit arbitrary – typically if this is to be done, it would be done in a jackknifing/bootstrapping way, in which the random samples are repeated many times to get a null distribution of expression values. I suggest choosing another approach that is more statistically sound, either the bootstrapping/jackknifing approach described or using e.g., DESeq2 or edgeR in R to test for statistically significant gene expression differences. See my major comments.

Fig. 2. I think the x-axis labels on panels a) and b) would be better labelled as Age (years) rather than Time (years). Also, the caption describes a solid red line but in the image the line is black, not red, and there are no dashed lines on a) and b) but rather coloured lines. Similarly the lines in c-f are described as dark solid lines but they are dashed. Also I am confused as to what this dashed line represents – it says the “average male-to-female fold change for SSD-associated genes”, but associated in which analyses? Clearly not the ones whose density is plotted (at least in c and e). Please revise the figure legend to accurately describe the images.

Fig. 3: I think the figure would be easier to read and interpret if the pink colour was removed. It is unnecessary as all species in the image are highlighted in pink.

Reviewer #2 (Remarks to the Author):

This is a revised version of the MS entitled “Sexual size dimorphism is associated with gene family contractions in brain development in mammals by Padilla-Morales et al, which analyzes the relationship of sexual size dimorphism (SDD) and gene family size. The authors conclude that SSD is associated with family sizes of genes related with brain development.

Although the MS has been extensively modified, the text remains confusing, and I find it very difficult to follow the authors' reasoning, which often leads me to different conclusions. For instance, they state that (lines 278-283) “...SSD-associated genes (...) and contracting SSD-associated genes show significantly higher expression in the brain (...) supporting the notion that these genes are likely to contribute to brain development functions”. With this sentence, as far as I understand, the authors suggest that SSD species with SSD-associated genes have more complex (and therefore, bigger) brains. They support this hypothesis in the next paragraph of the Discussion (lines 288-298): “Above enrichment analysis and transcriptional characterization (...) supporting a link between sexual selection and brain evolution”. In this paragraph, authors conclude that “...species with pair bonded mating systems have the largest brains...”. But the species with pair bonded mating systems (with big brains), usually are monogamous species, which have lower rates of SDD, and thereby, this reasoning goes against the initial hypothesis (i.e. SSD species have bigger brains), which is the hypothesis they were intended to support.

My general evaluation is that the article has not been substantially improved, and that it uses circular arguments, some of which lead to faulty reasoning.

RESPONSE TO REVIEWERS' COMMENTS

We appreciate the reviewers' comments. Our responses are in bold font, and sections of the manuscript are between double quotation marks (""") to support our answers. Line numbers refer to the revised manuscript.

Reviewer #1 (Remarks to the Author):

Major comments

The authors have clarified the focus of the manuscript and addressed many of my comments from the first round of review. I felt generally that the overall focus is much improved, although I have a number of suggestions for further improvement of the text – mainly, when topic sentences do not match the actual topics of the paragraphs. The improved flow of the manuscript led me to identify several concerns about the statistical analyses:

(Stats comment 1) The authors responded to my comment regarding removing gene families that did not expand or contract by saying that it did not change the results – this is a positive sign! However, the data file they shared would require a fair amount of work to interpret, so it is difficult for me to evaluate their assertion. I suggest including a supplementary figure – or, if it really makes no difference, include the dataset with all of the gene families included.

R = Thank you very much for this comment. We present the results of the PGLS SSD + BM but created by the no-variance dataset in the supplementary data 6. Also, we include in the same data file a list of all the overlapping gene families between the analyses, including the variance 0 and no-variance 0 in gene numbers. Moreover, you will find a Venn diagram showing those results in supplementary figure 4.

(Stats comment 2) The authors present the results of many phylogenetics least squares corrected regressions, including those of a model including just SSD, just body mass, and the joint effects of SSD and body mass. I am concerned by this – if the joint effects of the two factors are of interest, the model including both (and their interaction) should be presented and interpreted, not all three separate models. Analysing all three models could inflate type I error rates and seems somewhat misleading. I suggest the authors focus on the joint model and interpret the results from that model.

R = We really appreciated this notion, We focus our inferences on the bivariate model, including both SSD and log₁₀-transformed average body mass. Therefore, following the suggestion of this comment and comment 15 (see below), we have discarded the univariable model. We have specified this two-step approach in the MS.

(Stats comment 3) Gene expression profiles – comparing to 1000 random genes is not a particularly robust approach. Consider using proper bootstrapping/jackknifing analyses or implementing a statistical comparison of expression between tissues/ages in a package like DESeq2 or edgeR. **R = Thank you for**

this comment, we have used a bootstrapping approach to conduct the gene expression analyses as requested. See also comment 34.

In addition, the authors assume in a number of places that selection is the driving force behind gene family expansion or contraction, but one could expect some degree of gene family size change due to mutation and drift. The manuscript would benefit from ensuring that statements are not over-reaching, and that statements about selection (when not actually tested/modelled) are phrased as hypotheses for explaining the patterns arising from the data.

R= Thank you for this comment, we have changed the manuscript's language to avoid an over-reaching language.

When I first reviewed the manuscript, I had not realised the number of contracting gene families is actually larger than the number of gene family expansions. Do the authors have any intuition for whether contracting gene families are loss of whole segments of DNA or is it due to pseudogenisation of genes? This would be a useful addition to the discussion, in my opinion.

Specific comments

1. Abstract: Perhaps mention that more contractions occurred than expansions, potentially.

R= Thank you for the observation. We have included that in the abstract: “We reveal significant changes in gene family size linked to SSD. High SSD is associated with expanding gene families enriched in olfactory sensory perception functions. In contrast, the more numerous contracting gene families associated with brain development functions, which present high expression in the adult brain, were associated with reduced SSD.”

Lines 26 - 31

2. Lines 31-32: The authors make the assumption that contracting gene families are due to selective pressures, but isn't it equally possible that gene family contractions are due to a relaxation of selection pressures, and loss of genes due to mutation and drift?

R= Thank you for your observation. We have rewritten that section to mention that sexual selection is one of the factors contributing to gene family size changes but not the only one. “Our findings suggest that intense sexual selection serves as a facilitator of gene family expansion and contraction.” **lines 29 - 31**

3. Lines 53-62: The opening sentence of this paragraph focusses on mammals and birds, but then the focal example is in guppies (neither a mammal nor a bird). Consider re-framing this paragraph.

R= Thank you for your comment, the text with the new edits should be more direct and understandable. “Previous research has associated the evolution of various phenotypic traits with sexual size dimorphism in birds and mammals. Species with polygynous mating systems^{16,17} often exhibit higher male-biased SSD and higher species body mass, a pattern known as Rensch's rule¹⁶. Also, ornaments (such as fancy plumes and skin patterns, exaggerated tails) and armaments (i.e., antlers, enlarged fangs and spurs) indicate male-male competition¹⁸. Additionally, several mammalian lineages show an inverse correlation between male-biased SSD

and brain size^{11,19}. These associations highlight the potential role of sexual selection in shaping evolutionary trajectories. However, the complex relationship between sexual selection and other traits, especially brain size, remains understudied.” **lines 51 - 59**

4. Lines 70-72: Check this sentence – I believe there is unnecessary repetition of a phrase.

R= Thank you, we have removed that part of the introduction to avoid unnecessary repetition.

5. Lines 75-84: From this paragraph, I remain unclear on how comparative genomics offers insight into Rensch’s rule. Some of the information about sexual size dimorphism in this paragraph is also somewhat included in the first few paragraphs, so the authors might choose to delete this paragraph entirely, or merge it with for example paragraph 2 of the introduction.

R= Thank you, we agree with your observation, and that section is now removed.

6. Line 101: Consider rephrasing “gene families present a paradigm” as it was unclear what was meant by this statement.

R= Thanks for this observation, we have removed this section from the manuscript after implementing your other recommendations it seems redundant and does not have a real connection to our main focus.

7. Lines 105-106: In the previous work, did more duplications result in expanded olfactory capacity and ability to adapt to an increased variety of niches? Or was it a negative correlation and smaller gene families was associated with broader sets of traits? A clearer description of previous results related to gene family expansions/contractions will help the reader set expectations for your study.

R= Thank you for this comment, we have removed this section as well from the manuscript after following your other recommendations. it seems redundant and has no real connection to our main focus.

8. **Line** 108: Please specify what is being compared when mentioning ‘differences and similarities’

R= Thank you for this observation. We believe those sections do not add value to the introduction, so we opted to remove them from the manuscript.

9. **Lines** 109-111: This sentence is sufficiently vague that I do not know what the authors are encouraging the reader to consider.

R= Thank you for this comment, as mentioned in the comment before, we removed this section as it makes the text overly complex. Therefore, that section was removed.

10. Lines 115-116: The analysis of gene expression profiles was limited to humans, so was not really intricately interplaying with SSD in this paper. Consider re-phrasing to better reflect what was actually done in this manuscript.

R= Thank you for your comment, the new sentence should read like this: “Using functional annotations from humans, we then identified temporal expression patterns of genes within SSD-associated gene families in the

brain across prenatal and adult stages. Additionally, we examined the presence of sex-biased gene expression among these genes.” lines 87 - 90

11. Line 134: Change negatively to negative.

R= Thank you for noticing this mistake. Based on your suggestion in comment 15 and stats comment 2, this section was removed.

12. Line 135: Change less to fewer.

R= Thank you again for spotting this. This section was removed based on your suggestion in comment 15, and stats comment 2.

13. Lines 140-145: I suggest stating that body mass was log10-transformed here, given that the results now come before the methods.

R= Thank you, we agree with your comment. The new paragraph reads this way: “We utilised a phylogenetic generalised least square (PGLS) analysis, incorporating Benjamini-Hochberg correction³⁹, to assess the relationship between gene family size (as the dependent variable) and SSD, alongside log10-transformed average body mass (as independent variables).” lines 97 - 100

14. Lines 123-125: Consider more clearly stating how many gene families total were included in the analysis – I believe it was 5425, based on the n reported on line 125. Consider also re-iterating that this analysis included 124 species.

R= Thank you for the comment. We hope that now the paragraph is clearer than before. “We utilised a phylogenetic generalised least square (PGLS) analysis, incorporating Benjamini-Hochberg correction³⁹, to assess the relationship between gene family size (as the dependent variable) and SSD, alongside log10-transformed average body mass (as independent variables). A total of 5,425 gene families in 124 mammalian species were included into the analysis. The inclusion of body mass as separate variable in the model is supported by an allometric relationship between body mass and SSD, termed Rensch’s rule ¹⁶ ($r = 0.378$; $p < 0.001$; Supplementary Figure 1) ” lines 97 - 103

15. Lines 149-156: If the dual effects of SSD and body mass were of interest to the authors from the start, it seems unnecessary (and possibly inappropriate) to interpret results of both models that did not include both terms.

R= Thank you for pointing out this issue, We have addressed this section per your request. “We utilised a phylogenetic generalised least square (PGLS) analysis, incorporating Benjamini-Hochberg correction³⁹, to assess the relationship between gene family size (as the dependent variable) and SSD, alongside log10-transformed average body mass (as independent variables). A total of 5,425 gene families in 124 mammalian species were included into the analysis. The inclusion of body mass as separate variable in the model is supported by an allometric relationship between body mass and SSD, termed Rensch’s rule ¹⁶ ($r = 0.378$; $p < 0.001$; Supplementary Figure 1)

Our analysis revealed significant associations between gene family size expansion/contraction and SSD. Notably, the PGLS analysis with SSD corrected by log-transformed average body mass found a total of 340 SSD-associated gene families exhibiting statistically significant expansion ($p < 0.05$; effect size ranging from $r = 0.243$ to 0.674 ; figure 1a), whereas 405 showed statistically significant contraction ($p < 0.05$; effect size ranging from $r = -0.243$ to -

0.625; figure 1a). Furthermore, five gene families presented statistically significant contraction with log10-transformed average body mass corrected by SSD (effect size ranging from $r = -0.314$ to -0.407 ; Supplementary data 1). The only contracting gene family shared between log10-transformed average body mass and SSD (OG0000577) was associated with the homeobox genes (Entrez IDs: 30062, 84859, 30712), functioning in development and spermatogenesis. In the following sections, we will address genes within SSD-associated gene families as SSD-associated genes.” **lines 97 - 103 , 105 - 115.**

16. Lines 167-177: The encephalization index comes out of nowhere – encephalization should probably be introduced as a relevant topic in the introduction.

R= Thank you for your comment. We have introduced brain size and its relationship with SSD in the introduction. Authors use brain mass as brain size and transform it to relative brain size for their analyses (eg. <https://journals.plos.org/plosone/article?id=10.1371/journal.pone.0000062>, <https://www.ncbi.nlm.nih.gov/pmc/articles/PMC1762360/>). The encephalization index is calculated the same way as relative brain size therefore, we opt to use relative brain size instead. We hope this makes the manuscript more accurate and understandable for the public. “Additionally, several mammalian lineages show an inverse correlation between male-biased SSD and brain size^{10,16}.

Gene family expansion and contraction have essential roles influencing adaptive phenotypic diversity³³, as evidenced by the correlations with different biological functions^{34–36}, including the evolution of brain size and morphology^{30,31}.

Results section

Previous research has linked brain size with both SSD^{19,40} and gene family expansions during mammalian evolution³². To investigate this link, we examined gene family size in 57 species, for which data on relative brain size and SSD were available. Our results, correcting relative brain size with the SSD model, identified 290 gene families significantly associated with SSD, of which 51 showed gene family expansion ($p < 0.05$; effect size ranging from $r = 0.383$ to 0.581), and 239 showed contractions ($p < 0.05$; effect size ranging from $r = -0.382$ to -0.680 ; Figure 1a). Alternatively, the reverse correction identified 52 gene families. Among these, 25 were significantly under expansion ($p < 0.05$; effect size ranging from $r = 0.420$ to 0.677), and 27 were significantly contracted ($p < 0.05$; effect size ranging from $r = -0.430$ to -0.612 ; Supplementary data 1). There was minimal overlap between the gene families linked to SSD and brain size, with only one gene family showing a significant association with both traits ($p = 0.257$). This lack of significant overlap suggests that independent sets of genes influence these phenotypes.”

lines 55 - 57, 80 -83, 124 - 136.

17. Lines 168-169: This previous investigation into gene family expansions and brain size should definitely be featured in the introduction – this would be more relevant than the guppy example, probably.

R= Thank you for this comment. We hope now is more clearly displayed in the introduction. Placing this citation in the sexual selection and genome evolution paragraph would interrupt the flow of the introduction. Here is where this citation is included: “Additionally, several mammalian lineages show an inverse correlation between male-biased SSD and brain size^{10,16}

Gene family expansion and contraction have essential roles influencing adaptive phenotypic diversity³³, as evidenced by the correlations with different biological functions³⁴⁻³⁶, including the evolution of brain size and morphology^{30,31}. ” **lines 55 - 57, 80 - 83.**

18. Lines 197-198: I'm unclear on which analysis this sentence is referring to – is this the same model that was discussed on lines 138-148? Wouldn't this model have included all gene families, including those that both expanded and contracted? I think I am getting confused by the way the analyses are referred to in parentheses, but I think the authors need to clarify this for the results to be interpretable.

R= Thank you for highlighting the confusion, this model is represented as the PGLS that includes expanding and contracting gene families. After following your recommendations and the recommendations from comment number 15 and stats comment 2, we have removed the GO enrichments associated with one predictor PGLS. Now it reads this way: “Characterising the 340 SSD-associated expanded gene families found in the gene family expansion/contraction analysis, we observed 36 significantly enriched gene families (figure 1b). There was a significant overrepresentation of the sensory receptor of smell category, comprising 6.77% of the gene families significantly associated with the gene family expansion analysis (Figure 1b).

Of the 405 contracted SSD-associated gene families, 168 demonstrated significant enrichment (Figure 1b). Most of the enriched categories were associated with various aspects of development, including cell development (spinal cord development, skin development, muscle organ development, animal organ morphogenesis), general development (multicellular organism development) and brain development. Brain development-related processes represent 8.75% of the total contracting families, with significant overrepresentation in neuromuscular junction development ($p = 0.001$), neuron migration ($p = 0.002$), differentiation ($p < 0.001$) and forebrain development ($p = 0.006$) (Figure 1b).” **lines 139 - 151.**

19. Lines 210-211: Please articulate the number of samples from each tissue type at adult vs prenatal stages.

R= Thank you for your comment. That information can now be found on the third page of supplementary data 2. We opted not to include it in the text as our data set has a different number of samples between tissues.

20. Line 214: How was this random expectation of brain expression defined? Wouldn't you rather know whether these genes were up-regulated in adult brains relative to other adult tissues? And for contracting genes, what are 'randomly selected gene samples'? See my major comment re: statistical analysis.

R= Thank you for mentioning this. Yes, one of the things that we analyse is how brain genes associated with SSD are upregulated compared with other tissues. For that, we added a figure to be clearer (supplementary figure 2). Also, we are comparing upregulated genes between adults and fetal tissue separately. We rewrote the sentence for more clarity. Additionally, we have incorporated a bootstrapping approach into the analysis.

“Using human transcriptome expression profiles from 49 adult tissues and 20 from individuals in prenatal stages (Supplementary data 2), we tested whether expanding and contracting SSD-associated genes are predominantly expressed in the brain. Overall, brain expression had the highest rank for expanding and contracting genes in

adults and prenatal stages compared with other tissues, including the testis, which is known to have similar gene expression patterns with the brain³⁹ (Supplementary figure 2a-d). Nonetheless, in Adults, only the expression rank of contracting SSD-associated genes was significantly higher than expected by bootstrapping ($p < 0.001$; Supplementary figure 3b). For genes in prenatal stages, neither expanding nor contracting genes present an average rank for brain expression significantly higher compared to bootstrapped expectations ($p > 0.05$ in both datasets; Supplementary figure 3c-d).

...

We found eleven statistically significant genes for SSD-associated genes under expansion from prenatal individuals, with most presenting a negative fold change. Still, the gene 3670 was the most downregulated (Entrez ID = 3670, GO categories: Visceral motor neuron differentiation, spinal cord motor neuron cell fate speciation, pituitary gland development, neuron fate specification, negative regulation of neuron differentiation; Figure 2c). More than 25 statistically significant genes appear when contracting SSD-associated genes in prenatal individuals. Three of those genes are highly down-regulated (ENTREZ ID: 167826, 145258; log2 fold change < -2 ; Figure 2d). We identified twelve statistically significant genes for SSD-associated gene under expansion in adults; however, none exhibited high up or down-regulation. Nonetheless, categories such as neuron differentiation, forebrain development, central nervous brain development and pituitary gland development categories arise (Figure 2e). SSD-associated genes in adults presented more than 25 significant genes but only one with high overexpression (ENTREZ ID: 30062, GO category: hypothalamus development; log2 fold change < -2 ; Figure 2f). For all the SSD-associated genes with significant fold change, refer to supplementary data 3." **lines 156 - 165, 184 - 199.**

Supplementary figure 2:

21. Lines 219-235: It would be good to clarify in each of these sub-sections that these analyses are referring to humans only. I also encourage the authors to consider whether making inferences about sex biased expression patterns based only on human data is relevant to gene families across mammals, especially given that genes have generally high turnover in their sex bias (e.g., <https://doi.org/10.1126/science.aaw7317>, <https://doi.org/10.1126/science.aaw7317>). Are

there any gene families that expanded/contracted in humans (and/or primates) that could be a better focus for these interpretations?

R= Thank you for your very relevant comment. As you mentioned, some sex-biased expression patterns can be relevant for some mammals depending on tissue. In our research, we found sex-biased expressions of the SSD-associated genes expressed in the brain, but not enough to say that there is sex-biased gene expression linked to SSD-associated genes expressed in the brain.

Nonetheless, we added your suggestion to our text. “The high turnover of sex-biased genes makes it possible to infer sex-biased patterns across mammals based on human data²¹. The first approach showed no difference in sex-biased gene expression ($p > 0.05$) in the cortex, subcortex and cerebellum (Supplementary figure 4a-d). For our comparative analysis of expression levels, we categorised all SSD-associated genes under expansion and contraction by GO categories related to the human brain. Then, we selected statistically significant gene expression log₂ fold change values of each gene from prenatal individuals and adults. For SSD-associated genes under expansion from prenatal individuals, we found eleven statistically significant genes, with most of them presenting a negative fold change. Still, the gene 3670 was the most downregulated (Entrez ID = 3670, GO categories: Visceral motor neuron differentiation, spinal cord motor neuron cell fate specification, pituitary gland development, neuron fate specification, negative regulation of neuron differentiation; Figure 2c). Analysing SSD-associated genes under contraction in prenatal individuals, more than 25 statistically significant genes appear. Of those genes, three are highly down-regulated (ENTREZ ID: 167826, 145258; log₂ fold change < -2; Figure 2d). We identified twelve statistically significant genes for SSD-associated gene under expansion in adults; however, none exhibited high up or down-regulation. Nonetheless, categories such as neuron differentiation, forebrain development, central nervous brain development and pituitary gland development categories arise (Figure 2e). SSD-associated genes in adults presented more than 25 significant genes but only one with high overexpression (ENTREZ ID: 30062, GO category: hypothalamus development; log₂ fold change < -2; Figure 2f). For all the SSD-associated genes with significant fold change, refer to supplementary data 3..

...

Although thousands of genes have been identified as sex-biased in different species, and the high turnover in sex-biased genes makes it possible to create cross-species patterns, these assumptions should be made carefully, accounting for evolutionary history and genetic differences between species.” **lines 178 - 199, 271 - 275.**

22. Lines 244-246: The sentence about the associations being associated was very confusing, please rephrase.

R= Thank you for your comment. Now it reads this way “It is generally accepted that in mammals, male-biased SSD is driven by sexual selection, unlike female-biased SSD^{12,13,40}. The findings remained predominantly unaltered in analyses excluding 18 species, in which female size surpassed that of males. This suggests that interspecies male-biased sexual selection pressures likely influence the observed correlations.”

23. Lines 260-268: The GO associations for gene family expansions being related to gametes seems related to the similar research about mammalian testes size being related to the strength of postcopulatory sexual selection. I think the authors mention this somewhere else, but it seems as though it would be a relevant focus for this paragraph. Consider splitting the discussion of expanding gene family associations into its own paragraph from the contracting gene families.

R= Thank you for mentioning this. Following comment 15, we deleted the gene family size analysis with one variable (gene family size ~ SSD), so germ cell development is now not popping out in the GO enrichment analysis. Therefore, we removed that part of the discussion.

24. Lines 308-310: If gene families are contracting, I would think the members are less likely to have sex- and tissue-specific expression patterns – they will have more constraints/shared functions. Therefore I think this observation is not unexpected. I think this section could be improved by focussing on mechanisms and/or thinking about these patterns from this slightly more gene-centric view.

R= Thank you for this comment. As seen in comment 21, we have changed the sex-biased gene expression section to be more gene-centric. We analyse each gene and plot each significant fold change linked to brain function and development. The findings tend to show that a few of the SSD-associated genes expressed in the brain present sex bias.

Results section:

We found eleven statistically significant genes for SSD-associated genes under expansion from prenatal individuals, with most presenting a negative fold change. Still, the gene 3670 was the most downregulated (Entrez ID = 3670, GO categories: Visceral motor neuron differentiation, spinal cord motor neuron cell fate specification, pituitary gland development, neuron fate specification, negative regulation of neuron differentiation; Figure 2c). More than 25 statistically significant genes appear when contracting SSD-associated genes in prenatal individuals. Three of those genes are highly down-regulated (ENTREZ ID: 167826, 145258; log₂ fold change < -2; Figure 2d). We identified twelve statistically significant genes for SSD-associated gene under expansion in adults; however, none exhibited high up or down-regulation. Nonetheless, categories such as neuron differentiation, forebrain development, central nervous brain development and pituitary gland development categories arise (Figure 2e). SSD-associated genes in adults presented more than 25 significant genes but only one with high overexpression (ENTREZ ID: 30062, GO category: hypothalamus development; log₂ fold change < -2; Figure 2f). For all the SSD-associated genes with significant fold change, refer to supplementary data 3.

Discussion section:

This study reveals that within the SSD-associated genes expressed in the brain undergoing expansion and contraction, there are marginal differences in sex-biased gene expression, with four genes highly downregulated in females and one upregulated in females compared to males. Although thousands of genes have been identified as sex-biased in different species, and the high turnover in sex-biased genes makes it possible to create cross-species patterns, these assumptions should be made carefully, accounting for evolutionary history and genetic differences between species^{20,72} **lines 184 - 199, 269 - 275.**

25. Lines 312-313: More than references 41 and 42 have studied transcriptome data from multiple species. For example, see the following (and there are probably more):

<https://doi.org/10.1111/mec.15115>

<https://doi.org/10.1126/science.aaw7317>

<https://doi.org/10.1126/science.aaw7317>

<https://doi.org/10.1126/science.adf1046>

<https://doi.org/10.1111/evo.14579>

<https://doi.org/10.1111/mec.13596>

<https://doi.org/10.1073/pnas.1501339112>

R= Thank you for sharing those references with us. That sentence and the part of the paragraph after it are no longer needed, but those references were used to support other sections of the text.

26. Lines 313-314: Re-emphasising the brain results here seems out of place – consider moving to the other sections where the brain results are discussed.

R= Thank you for this comment. That section was removed from the text after the changes it is not relevant anymore.

27. Lines 315-326: The link from the results in this paper to this example by Tripp et al was not very clear and I felt detracted from the key messages of the manuscript.

R= Thank you for noticing it. As you said, we removed that example because it might be distracting for the reader.

28. Lines 371-373: I thought that only 124 species were retained because that was the number that had available body size data. Please clarify the number of species used in downstream analyses.

R= Thank you for your comment. Yes, 124 species were used in the PGLS analysis because of the availability of body mass and SSD. But orthology annotations were done for 142 species. “A phylogenetic tree containing 142 of the 146 of the above-described set of species was downloaded (16/08/20) from Timetree⁸⁶; the not overlapping species between the phylogenetic tree and the list of species with an available genome were excluded from further analyses.

Gene family annotation

Gene families for 142 species were annotated using Orthofinder⁸⁷. Initially, we selected the longest available CDS sequence for each gene. Subsequently, all remaining CDS sequences within and between species were aligned with “DIAMOND”⁸⁸, as it noun for its speed, high sensitivity and scalability needed to handle large datasets⁸⁷. Orthologue gene groups were constructed utilising a predefined phylogenetic tree, as described above.

...

Subsequently, we conducted a two-predictor PGLS^{92,93} to examine associations between gene family size and the focal traits across 124 species (gene family size ~ SSD + log10-transformed average body mass) for which gene family annotations and phenotype data were available.” **lines 326 - 329, 332 - 336, 349 - 352.**

29. Lines 385-386: I suggest rephrasing to “The same gene families were associated with SSD in an analysis where all gene families were included (Supplementary data 5)”. Although, this would be better represented as a supplementary figure (perhaps instead of the data file, or in addition to) – the file is difficult to interpret and compare easily to the main results presented, so it is difficult for me to evaluate whether my concern is addressed.

R= Thank you for your suggestion, we incorporated your recommendation into the text. Additionally, in the second sheet of the supplementary data 6 you will find the list of the gene families from both tests (Gene families with and without variation in gene number). We added a Venn diagram showing the overlap. “Additionally, gene families with no variation in gene number across species were removed from the analysis, as regression analyses aim to construct models to explain significant amounts of

variance. The same gene families were associated with SSD in an analysis where all gene families were included (Supplementary data 6; Supplementary figure 5). ” lines 346 - 349.

Supplementary figure 5

30. Lines 400-401: I'm confused about which analysis the methodology was based off of, and whether encephalization and SSD were included in this analysis.

R= Thank you for noticing this issue, we have changed that sentence and now reads like this:

“Furthermore, we conducted a similar analysis to assess the relationship between gene family expansion and relative brain size (gene family size \sim SSD + Relative brain size), following the same methodology described above used for examining the association between gene family expansions and contractions with SSD and log₁₀-transformed average body mass. This analysis was performed on a subset of 57 species for which both relative brain size and SSD data were available (Supplementary data 7). Relative brain size was calculated using brain size relative to body mass ($\log(\text{relative brain size}/\text{body mass})$), while accounting for the allometric effect of body size, by calculating residuals of a log-log least squares linear regression of brain size against body mass⁹². ” lines 356 - 364.

31. Lines 405-406: I am confused about which variables went into this model, please rephrase. Was body size or SSD included as explanatory/random variables, or was it just the encephalization index?

R= Thank you for letting us this. This was done to calculate SSD corrected by relative brain size (previously called encephalization index). Now the section should read more clearly: “Furthermore, we conducted a similar analysis to assess the relationship between gene family expansion and relative brain size (gene family size \sim SSD + Relative brain size), following the same methodology described above used for examining the association between gene family expansions and contractions with SSD and log₁₀-transformed average body mass. This analysis was performed on a subset of 57 species for which both relative brain size and SSD data were available (Supplementary data 7). Relative brain size was calculated using brain size relative to body mass ($\log(\text{relative brain size}/\text{body mass})$), while accounting for the allometric effect of body size, by calculating residuals of a log-log least squares linear regression of brain size against body mass⁹². ” lines 356 - 364.

32. Line 427: Here in the methods it states that 178 tissue samples were sourced, but the methods describes using 49 adult and 20 prenatal tissues. Please clarify in the main text.

R= Thank you for this comment: the new version should clearly point out that the 178 tissues were averaged, resulting in 49 adult and 20 prenatal tissue samples. “Then to conduct the gene expression ranking analysis in prenatal and adult stages, we used the SSD-associated genes from the gene families obtained in

the SSD + log10-transformed average body mass PGLS. Human transcriptome data for 178 tissue samples were sourced from the Fantom database release 5⁹¹. Gene expression levels were averaged, from the original 178 tissues, for samples corresponding to brain areas and other tissues resulting in 49 adult and 20 prenatal healthy tissues, including the brain (Supplementary data 2). " **lines 395 - 400.**

33. Lines 430-432: Why was gene expression in the central nervous system used as the point of comparison for the brain expression?

R= Thank you for this clarification. We used the brain as a focal trait, so it has to be written as brain and not central nervous system, it was a mistake on our part to use the wrong term. "For each gene, the number of tissues with gene expression levels higher than in the brain was calculated and averaged for SSD-associated genes." **lines 400 - 402.**

34. Lines 433-435: Comparing the expression to 1000 random samples seems a bit arbitrary – typically if this is to be done, it would be done in a jackknifing/bootstrapping way, in which the random samples are repeated many times to get a null distribution of expression values. I suggest choosing another approach that is more statistically sound, either the bootstrapping/jackknifing approach described or using e.g., DESeq2 or edgeR in R to test for statistically significant gene expression differences. See my major comments.

R= Thank you for your comment there. We have addressed this and now include a bootstrapping approach. The result follows the same trend. Methods section: "The statistical significance of gene expression in the brain was evaluated using a bootstrapping approach. We generated 10,000 bootstrap samples by randomly selecting the same number of genes from the dataset with replacement. Prenatal and adult tissues were processed separately." **lines 402 - 405.**

35. Fig. 2. I think the x-axis labels on panels a) and b) would be better labelled as Age (years) rather than Time (years). Also, the caption describes a solid red line but in the image the line is black, not red, and there are no dashed lines on a) and b) but rather coloured lines. Similarly the lines in c-f are described as dark solid lines but they are dashed. Also I am confused as to what this dashed line represents – it says the "average male-to-female fold change for SSD-associated genes", but associated in which analyses? Clearly not the ones whose density is plotted (at least in c and e). Please revise the figure legend to accurately describe the images.

R= Thank you for correcting this issue. Figure 2 was changed to be more informative and be more gene-centric. New legend: "Panels show average gene expression levels for A) SSD-associated genes undergoing expansion on average for humans and (slope 0.08) B) SSD-associated genes undergoing contraction on average for humans (slope -0.08). The solid dark line represents when gestation time finishes (nine months). Gene expression data was log2(x+1) transformed to account for zero expression values and mitigate the impact of low expression levels. Panels showing statistically significant female-to-male gene expression log2 fold change compared to GO categories in C) prenatal brain for expanding SSD-associated genes, D) prenatal brain for contracting SSD-associated genes, E) adult brain for expanding SSD-associated genes and F) adult brain for contracting SSD-associated genes. The dashed dark line marks the log2 fold change of 2 and -2, representing when fold change values are twice as large as those observed in the other phenotype. " **lines 447 - 455.**

36. Fig. 3: I think the figure would be easier to read and interpret if the pink colour was removed. It is unnecessary as all species in the image are highlighted in pink.

R= Thank you for your comment. The colour was removed. Line 456

Reviewer #2 (Remarks to the Author):

1. This is a revised version of the MS entitled "Sexual size dimorphism is associated with gene family contractions in brain development in mammals by Padilla-Morales et al, which analyzes

the relationship of sexual size dimorphism (SSD) and gene family size. The authors conclude that SSD is associated with family sizes of genes related with brain development.

2. Although the MS has been extensively modified, the text remains confusing, and I find it very difficult to follow the authors' reasoning, which often leads me to different conclusions. For instance, they state that (lines 278-283) "...SSD-associated genes (...) and contracting SSD-associated genes show significantly higher expression in the brain (...) supporting the notion that these genes are likely to contribute to brain development functions". With this sentence, as far as I understand, the authors suggest that SSD species with SSD-associated genes have more complex (and therefore, bigger) brains. They support this hypothesis in the next paragraph of the Discussion (lines 288-298): "Above enrichment analysis and transcriptional characterization (...) supporting a link between sexual selection and brain evolution". In this paragraph, authors conclude that "...species with pair bonded mating systems have the largest brains...". But the species with pair bonded mating systems (with big brains), usually are monogamous species, which have lower rates of SSD, and thereby, this reasoning goes against the initial hypothesis (i.e. SSD species have bigger brains), which is the hypothesis they were intended to support.

My general evaluation is that the article has not been substantially improved, and that it uses circular arguments, some of which lead to faulty reasoning.

R= Thank you for your valuable feedback. Now, the narrative and objectives of the manuscript are clear, and should not be contradictory. The paper's premise is that SSD influences gene family expansion and contraction. The genes within gene families under contraction are linked with brain development and function. Therefore, species with higher SSD are having a gene family contraction in genes related with brain development. Now, in our introduction and discussion, we present that species with lower dimorphism present bigger brains and more complex behaviors.

In this paragraph, we present that SSD presents an inverse correlation with brain size.

"Previous research has associated the evolution of various phenotypic traits with sexual size dimorphism in birds and mammals. Species with polygynous mating systems^{16,17} often exhibit higher male-biased SSD and higher species body mass, a pattern known as Rensch's rule¹⁶. Also, ornaments (such as fancy plumes and skin patterns, exaggerated tails) and armaments (i.e., antlers, enlarged fangs and spurs) indicate male-male competition¹⁸. Additionally, several mammalian lineages show an inverse correlation between male-biased SSD and brain size^{11,19}. These associations highlight the potential role of sexual selection in shaping evolutionary trajectories. However, the complex relationship between sexual selection and other traits, especially brain size, remains understudied." **lines 51 - 59.**

In this results section, we present the SSD-associated genes and their enrichment in brain development functions

"Enrichment of SSD-associated genes in brain and olfactory functions

...

Of the 405 contracted SSD-associated gene families, 168 demonstrated significant enrichment (Figure 1b). Most of the enriched categories were associated with various aspects of development, including cell development (spinal

cord development, skin development, muscle organ development, animal organ morphogenesis), general development (multicellular organism development) and brain development. Brain development-related processes represent 8.75% of the total contracting families, with significant overrepresentation in neuromuscular junction development ($p = 0.001$), neuron migration ($p = 0.002$), differentiation ($p < 0.001$) and forebrain development ($p = 0.006$) (Figure 1b).” **lines 138 - 151.**

Here, we present that complex behaviors like parental care are present in species with a tendency to monomorphism and also present big brains

“The enrichment analysis and transcriptional characterisation presented above provide compelling evidence for an evolutionary link between sexual selection and brain evolution. In mammals, the evolution of SSD is mainly attributed to selective pressures acting on male body size in the context of sexual selection^{3,67}. Polygynous populations, where larger males outcompete smaller ones in competition for access to multiple mates⁶⁸, exhibit higher rates of SSD. In contrast, monogamous species, often characterised by biparental care, typically display lower rates of SSD^{37,69}. Such behavioural phenotypes accompanying monogamous mating systems likely result from changes in complex social skills leading to changes in brain function and size (see Schillaci 2006³⁴ for an example in primates). In birds and mammals, species with pair-bonded mating systems present the largest brains^{34,70,71}, further supporting a link between sexual selection and brain evolution.” **lines 258 - 268.**

REVIEWERS' COMMENTS

Reviewer #1 (Remarks to the Author):

The paper is much easier to follow in its revised form and the statistical analyses are appropriate. I commend the authors for their hard work on this. I have only three remaining major comments:

- (1) The statistical analysis is focussed on modelling gene family size, but in the results and discussion genes are described as falling into the discrete categories of expanding, contracting, or (presumably) staying the same. However, the conversion from the model of gene family size to these discrete categories was not clearly articulated and needs to be included in the methods.
- (2) I feel that the authors overstate the link between sexual selection and gene family expansion. For example, in the abstract, the authors state that their "findings suggest that intense sexual selection serves as a facilitator of gene family expansion and contraction." Although sexual size dimorphism is linked to gene family expansions in their results, I do not think the authors have reason to state such a strong causal relationship. In addition, it's an indirect link, as the analysis is focussed on sexual size dimorphism. Furthermore, the relationships between brain development and gene family size actually indicate that other selective pressures – not sexual selection – are likely highly important in driving the evolution of gene family size. I think the authors need to provide more nuance in their discussion of the link between their results and sexual selection – or indeed other forms of selection, as selection was not measured at any point in the analysis. Providing more nuance will be important in the abstract and discussion in particular.
- (3) The order of the figures is somewhat odd – Fig. 3 would make most sense as the first figure, I would think.

Specific comments:

Lines 26-31: The new text in the abstract aims to clarify the results but I found was difficult to follow and potentially could be misleading – I'm not sure the authors have done any analysis of sexual selection so it is a bit of a reach to state that intense sexual selection facilitates gene family expansion and contraction. Indeed, the association between brain functions, reduced SSD, and contracting gene families would be indicative of selective pressures other than sexual selection having a primary role. See my major comment above.

Line 62: over-representation of sex-biased genes on sex chromosomes: other research might also be useful to look at other than just the Meisel et al paper. Also, I'm not sure sex chromosomes need to play a prominent role in your introduction, given the paper's focus. Were gene families with members on sex chromosomes included in the analyses, or were sex chromosome genes excluded? I do not believe this was stated or mentioned in the discussion.

Lines 178-179, "The high turnover..": I'm not sure I understand this logic. Wouldn't high turnover make it challenging to extrapolate about sex-biased patterns across mammals based only on human data?

Line 179: "The first approach": What is the first approach? I'm not sure I follow what is being described here.

Lines 208-209: These three citations do not convince me, especially as the third one is about fecundity selection, not sexual selection.

Lines 276-278: I'm not sure that 'detrimental selection' is the best phrase. Smaller SSD is associated with gene family contractions, that is not necessarily detrimental selection. And there is no evidence that this is related to sexual selection. I think the authors are attempting to imply causal relationships from a correlative analysis.

Line 334: 'noun' should be 'known'

Line 357: This is not gene family expansion, it is just gene family size.

Lines 361-364: The calculation of relative brain size is unclear. Was it the log(brain size/body mass) or the residuals from the log-log least squares linear regression?

Line 385: Were there 18947 genes that were significantly associated with the SSD + body mass PGLS?

Fig 2 caption: Log2 fold change of 2 is 4x as large, not 2x as large as stated on lines 454-455.

Reviewer #1 (Remarks on code availability):

I have briefly glanced over the code and data but have not done a thorough review or checked that the code reproduces the results/figures presented in the manuscript. The code repository has a complete README file and is organised in a way that is easy to interpret.

RESPONSE TO REVIEWERS' COMMENTS

We appreciate all the comments given to us, all of the are helpful to increase the quality of the paper. Our responses are in bold font, and sections of the manuscript are between double quotation marks (“”) to support our answers. Line numbers refer to the revised manuscript.

Reviewer #1 (Remarks to the Author):

The paper is much easier to follow in its revised form and the statistical analyses are appropriate. I commend the authors for their hard work on this. I have only three remaining major comments:

(1) The statistical analysis is focussed on modelling gene family size, but in the results and discussion genes are described as falling into the discrete categories of expanding, contracting, or (presumably) staying the same. However, the conversion from the model of gene family size to these discrete categories was not clearly articulated and needs to be included in the methods.

R= Thank you for your observation. We further explained how we determined the expanding and contracting gene families. Lines 364- 381

“Gene family size analyses

To identify gene families associated with SSD, to be included in the analysis gene families were required to be present in at least 80% of species to filter out those which are lineage-specific, have at least three genes in at least one species to avoid gene presence-absence comparisons. Additionally, gene families with no variation in gene number across species were removed from the analysis, as regression analyses aim to construct models to explain significant amounts of variance. The same gene families were associated with SSD in an analysis that included all gene families (Supplementary data 6; supplementary figure 5).

Using the selected gene counts per gene family per species, we conducted a two-predictor PGLS^{99,100} to examine associations between gene family size and the focal traits across 124 species (gene family size ~ SSD + log₁₀-transformed average body mass) for which gene family annotations and phenotype data were available. We employed the “nlme” v.3.1-152 R package¹⁰¹, assuming a Brownian motion model of evolution. Post PGLS execution, we calculated *r* values from *t* values and adjusted *p*-values for multiple testing using the Benjamini-Hochberg correction.

Gene families presenting a significant positive correlating with gene family size ($r > 0.3$), where catalogued as expanding gene families. While contracting gene families were denominated when presenting a significant negative correlation with gene family size ($r < -0.3$). ”

(2) I feel that the authors overstate the link between sexual selection and gene family expansion. For example, in the abstract, the authors state that their “findings suggest that intense sexual selection serves as a facilitator of gene family expansion and contraction.” Although sexual size dimorphism is linked to gene family expansions in their results, I do not think the authors have reason to state such a strong causal relationship. In addition, it’s an indirect link, as the analysis is focussed on sexual size dimorphism. Furthermore, the relationships between brain development and gene family size actually indicate that other

selective pressures – not sexual selection – are likely highly important in driving the evolution of gene family size. I think the authors need to provide more nuance in their discussion of the link between their results and sexual selection – or indeed other forms of selection, as selection was not measured at any point in the analysis. Providing more nuance will be important in the abstract and discussion in particular.

R= Thank you for this observation. We have reviewed the text and removed the language that could be indicating causality towards sexual selection only. We have added some new sections in the discussion addressing the fact that sexual size dimorphism and brain evolution have evolved by natural selection and not only by sexual selection. Lines 278-283, 306-313 “However, natural and sexual selection are not necessarily mutually exclusive^{43,80,81}. Ecological selection pressures may also significantly contribute to sexual size dimorphism⁸² through competitive displacement, binomial and dimorphic niches⁸³. Evidence suggests that in some clades, such as pinnipeds, sexual size dimorphism arose prior to the emergence of polygyny^{84,85}, showing the multifactor nature in the origin of complex phenotypes such as sexual size dimorphism.

...

The evolution of the mammalian brain can follow different pathways as it directs an animal’s interaction with its environment⁹⁰. The gene family size changes observed in this work may be influenced by shared ecological pressures alongside sexual selection. For instance, the instrumental hypothesis states that species develop larger brains to forage more efficiently, while the social hypothesis suggests that large brains have evolved for the need to gain key social skills⁹¹. Investigating the interplay of the social taxa proportion, foraging strategies, and parental care in relation to sexual size dimorphism and gene family size changes could provide a better understanding of genome evolution dynamics.”

(3) The order of the figures is somewhat odd – Fig. 3 would make most sense as the first figure, I would think.

R= Thank you for this edit. Now the figure 3 is the figure 1. Line 460

Specific comments:

1. Lines 26-31: The new text in the abstract aims to clarify the results but I found was difficult to follow and potentially could be misleading – I’m not sure the authors have done any analysis of sexual selection so it is a bit of a reach to state that intense sexual selection facilitates gene family expansion and contraction. Indeed, the association between brain functions, reduced SSD, and contracting gene families would be indicative of selective pressures other than sexual selection having a primary role. See my major comment above.

R= Thank you for your observation. The abstract was re-written and now it reads this way Lines 25-36 “In mammals, sexual size dimorphism often reflects the intensity of sexual selection, yet its connection to genomic evolution remains unexplored. Gene family size evolution can reflect shifts in the relative importance of different molecular functions. Here, we investigate the association between brain development gene repertoire to sexual size dimorphism using 124 mammalian species. We reveal significant changes in gene family size linked to sexual size dimorphism. High levels of dimorphism correlate with an expansion of gene families enriched in olfactory sensory perception and a contraction of

gene families associated with brain development functions, many of which exhibited particularly high expression in the human adult brain. These findings suggest a relationship between intense sexual selection and alterations in gene family size. These insights illustrate the complex interplay between sexual dimorphism, gene family size evolution, and their roles in mammalian brain development and function, offering a valuable understanding of mammalian genome evolution.”

2. Line 62: over-representation of sex-biased genes on sex chromosomes: other research might also be useful to look at other than just the Meisel et al paper. Also, I’m not sure sex chromosomes need to play a prominent role in your introduction, given the paper’s focus. Were gene families with members on sex chromosomes included in the analyses, or were sex chromosome genes excluded? I do not believe this was stated or mentioned in the discussion.

R= Thank you for this comment. As you said, this is not part of our main focus. Therefore, it would be better to remove it from the introduction. Line 67

3. Lines 178-179, “The high turnover...”: I’m not sure I understand this logic. Wouldn’t high turnover make it challenging to extrapolate about sex-biased patterns across mammals based only on human data?

R= Thank you for your comment. Yes, you are right; removing that mistake was done to improve the uncertainty of the text, and now it reads this way. Line 184-185 “Although there is a relatively high turnover of sex-biased genes²², if SSD-associated genes tend to be sex-biased, we may expect a significantly higher degree of sex-biased gene expression among SSD-associated genes based on gene expression human data.”

4. Line 179: “The first approach”: What is the first approach? I’m not sure I follow what is being described here.

R= Thank you for observing this confusing beginning of the sentence. The new sentence reads this way. Lines 187-189 “Temporal patterns of cortical expression showed no difference in sex-biased gene expression ($p > 0.05$) in the cortex, subcortex and cerebellum (Supplementary figure 4a-d).”

5. Lines 208-209: These three citations do not convince me, especially as the third one is about fecundity selection, not sexual selection.

R= Thank you for raising this misconception. Now, we rephrased this sentence to better deliver our message. Also, we have added more references to mention the classic association of SSD and sexual selection in mammals and the fecundity selection association with female-biased SSD. Lines 216-219 “It is commonly accepted that in mammals, male-biased SSD is primarily driven by sexual selection⁴⁶⁻⁵⁰ acting on males, while the role of sexual selection in the evolution of female-biased SSD has been more debated, though not entirely ruled out unlike female-biased SSD which is commonly attributed to fecundity selection^{13,47,51}.”

6. Lines 276-278: I’m not sure that ‘detrimental selection’ is the best phrase. Smaller SSD is associated with gene family contractions, that is not necessarily detrimental selection. And there is no evidence that this is related to sexual selection. I think the authors are attempting to imply causal relationships from a correlative analysis.

R= Thank you for your good observation. We have rephased the paragraph, and it now reads this way. Lines 291-294 “Our findings illustrate potential selection pressures affecting mammalian brain evolution in the context of SSD, often used as a proxy of sexual selection. This perspective potentially differs from Geoffrey Miller's "Mating Mind Hypothesis", which focuses on the evolution of human cognitive and behavioural traits through sexual selection⁷⁹.”

7. Line 334: ‘noun’ should be ‘known’

R= Thank you for spotting this mistake. We have fixed it, and now it reads “known”. Line 356

8. Line 357: This is not gene family expansion, it is just gene family size.

R= Thank you for remarking on this typo. We have changed it to gene family size. Line 383

9. Lines 361-364: The calculation of relative brain size is unclear. Was it the log(brain size/body mass) or the residuals from the log-log least squares linear regression?

R= Thank you for noticing this, and sorry for this mistake. We have removed the mistaken part, and now it reads this way. Lines 387 - 389 “Relative brain size was calculated using brain size controlled by the allometric effect of body mass, by calculating the residuals of a log-log least squares linear regression of brain size against body mass⁹⁵”

10. Line 385: Were there 18947 genes that were significantly associated with the SSD + body mass PGLS?

R= Thank you for this question; now the text will read more accurately. Lines 410 – 412 “Gene expression data for 18,947 protein-coding genes were downloaded from BrainSpan⁹⁷, with 933 of the genes overlapping with our SSD + body mass PGLS dataset,”

11. Fig 2 caption: Log2 fold change of 2 is 4x as large, not 2x as large as stated on lines 454-455.

R= Thank you for this clarification. The change was done. Line 480 “The dashed dark line marks the log2 fold change of 2 and -2, representing when fold change values are four times as large as those observed in the other phenotype.”

Reviewer #1 (Remarks on code availability):

12. I have briefly glanced over the code and data but have not done a thorough review or checked that the code reproduces the results/figures presented in the manuscript. The code repository has a complete README file and is organised in a way that is easy to interpret.

R= Thank you for your comment.